# OPEN-SET SEMANTIC GAUSSIAN SPLATTING SLAM WITH EXPANDABLE REPRESENTATION

**Yucheng Yan**[1,2]**, Chen Liang**[1]**, Wenguan Wang**[1]***& Yi Yang**[1]

[1]The State Key Lab of Brain-Machine Intelligence, Zhejiang University
[2]Shanghai Innovation Institute

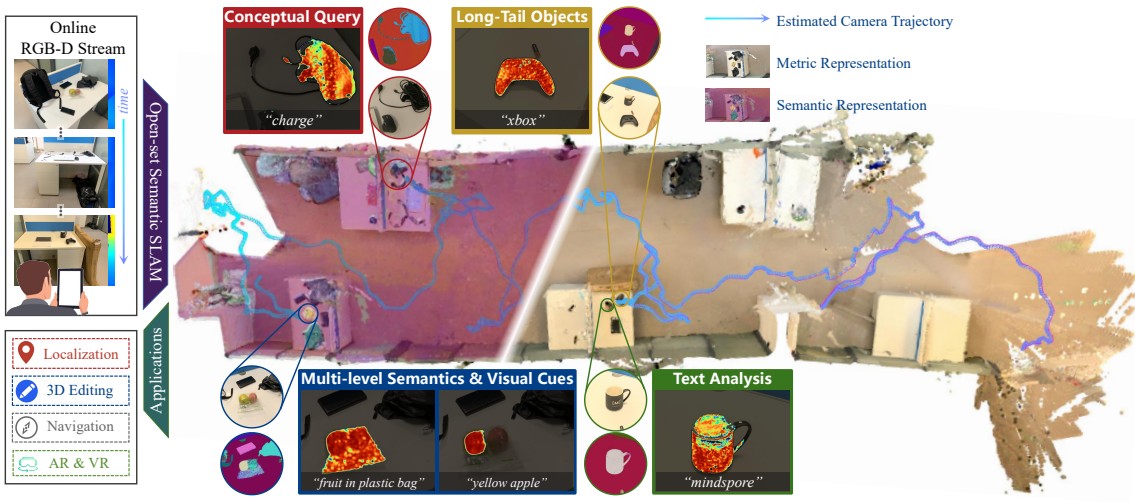

Figure 1: This work introduces Open-Set Semantic Gaussian Splatting SLAM, a system designed to enable everyday devices (*e.g.*, smartphones) to capture and reconstruct in-the-wild 3D scenes with rich, open-set semantics on top of SLAM frameworks.

## ABSTRACT

This work enables everyday devices, *e.g.*, smartphones, to dynamically capture open-ended 3D scenes with rich, expandable semantics for immersive virtual worlds. While 3DGS and foundation models hold promise for semantic scene understanding, existing solutions suffer from unscalable semantic integration, prohibitive memory costs, and cross-view inconsistency. To respond, we propose Open-Set Semantic Gaussian Splatting SLAM, a GS-SLAM system augmented by an expandable semantic feature pool that decouples condensed scene-level semantics from individual 3D Gaussians. Each Gaussian references semantics via a lightweight indexing vector, reducing memory overhead by orders of magnitude while supporting dynamic updates. Besides, we introduce a consistency-aware optimization strategy alongside a Semantic Stability Guidance mechanism to enhance long-term, cross-view semantic consistency and resolve inconsistencies. Experiments demonstrate that our system achieves high-fidelity rendering with scalable, open-set semantics across both controlled and in-the-wild environments, supporting applications like 3D localization and scene editing. These results mark an initial yet solid step towards high-quality, expressive, and accessible 3D virtual world modeling. Our code will be released on our Project Page.

## 1 INTRODUCTION

A 3D virtual world functions as a collective space where user avatars interact seamlessly within a metric-semantic representation of 3D environments, capturing both appearance and semantics [1, 2].

---

*Corresponding author

Recent technological strides in platforms, such as VisionPro [3] and Metaverse [4], have driven the transition toward richer content, larger-scale environments, and the ambitious goal of digitizing the entire Earth [5–9]. However, achieving this vision currently requires specialized and expensive hardware [10], limiting scalability and accessibility. This work aims to bridge this gap by enabling everyday devices, such as handheld smartphones, to capture in-the-wild 3D scenes while identifying a broad spectrum of semantics (*c.f.*, Fig. 1), including multi-level semantics and visual properties ('*fruit in plastic bag*', '*yellow apple*'), long-tail objects ('*xbox*'), *etc.*, to support diverse applications in open-ended 3D virtual worlds.

One natural solution is to distill open-set semantics from 2D foundation models like CLIP [11] or SAM [12] into 3DGS-based representations [13], then optimizing them within a simultaneous localization and mapping (SLAM) framework. Early efforts along this direction generally follow two strategies for semantic integration: *(i)* Directly encoding feature vectors into each 3D Gaussian, alongside color attributes, to represent semantics [14, 15]. *(ii)* Assigning pixel-wise semantics after rendering the scene into 2D maps, bypassing explicit 3D point-wise semantics [16, 17]. While promising, these approaches struggle to scale as environments gradually evolve as in SLAM: ❶ Both methods are trained to fit pre-extracted semantic features of a fixed scene, making them incapable of accommodating new concepts dynamically. ❷ For *(i)*, memory consumption grows prohibitively large as the number of Gaussians and feature dimensions increase. Existing solutions often compress feature dimensions [14], *i.e.*, 3 *vs.* 512 (original CLIP dimension), to reduce costs, sacrificing semantic expressiveness. ❸ For *(ii)*, semantics are only recoverable on 2D maps, whereas applications such as 3D localization, embodied navigation, and scene editing require fully 3D-aware semantics. ❹ Due to 2D nature of foundation models, ensuring semantic coherence across multiple viewpoints is challenging, hindering effective 3D semantic fields learning.

Focusing on the aforementioned challenges, we propose Open-Set Semantic Gaussian Splatting SLAM. Our system in brief is a GS-SLAM system augmented by an expandable representation of open-set semantics. At its core, the system maintain an expandable semantic feature pool that stores condensed semantics across the entire scene. This pool can dynamically incorporate incoming semantics, updating existing ones, and increase its capacity to accommodate continuously evolving novel concepts. Each 3D Gaussian is associated with a low-dimensional indexing vector that retrieves relevant semantics from this shared pool. This design offers four advantages: First, its expandable nature allows for continuous integration of new semantic concepts, *addressing issue one*. Second, the shared pool by design is significantly smaller than the number of Gaussian points, *i.e.*, 200 *vs.* $5 \times 10^6$, reducing memory overhead and enabling scalability (*issue two*). Third, each Gaussian explicitly aggregates 3D point-specific semantics (*issue three*). Besides, fourth, since local Gaussians representing the same object should exhibit similar semantics, this deign diminishes local semantic redundancy and promotes field learning.

Moreover, to enhance semantic coherence (*issue four*), we implement a consistency-aware optimization strategy ensuring that semantic representations align both within frames (*intra*-frame) and across frames (*inter*-frame). We employ contrastive learning to enforce these correspondences and introduce an Intra-Inter Semantic Consistency Objective during semantic field learning. Additionally, we introduce a Semantic Stability Guidance mechanism to mitigate semantic ambiguity, where different semantics may be attributed to the same object. By measuring inter-frames pixel-to-object semantic consistency through cosine similarity, we adjust the learning signal, reducing influence from inconsistent regions while amplifying signals from consistent regions, thereby enhancing the overall coherence and stability of semantic representations within our system.

For examination, we compare our framework with both NeRF-based [18] and 3DGS-based [19, 20] SLAM methods on Replica [21]. Experimental results demonstrate that our system establishes a precise metric-semantic field. As seen in §4.2, we enhance current SLAM frameworks on rendering quality and tracking accuracy with **1.32 – 3.50** PSNR gains and over **0.07** ATE RMSE reductions. In §4.3, we show that our framework accurately capture closed-set object categories (**0.13% – 9.54%** mIoU improvements). In §4.4, our framework showcases the ability of 3D explicit open-set semantic reconstruction, facilitating 2D segmentation (over **7.2%** mIoU improvements), 3D object localization and 3D scene editing. Notably, our method is applicable on in-the-wild scenes captured by hand-held everyday devices and able to construct an open-set semantic representation (§4.4).

## 2 RELATED WORK

**3D Scene Representation.** For a long time, point clouds have served as one of the most widely used representations for 3D scenes, due to their simplicity and compatibility with classical reconstruction pipelines. Several studies [22, 23] have further attempted to enrich point clouds with semantic segmentation and object-level understanding. However, most of these approaches focus on interpreting pre-reconstructed point clouds, rather than jointly optimizing both geometry and semantics directly from multi-view images. Recently, NeRFs [24] have emerged as a pivotal technique, enabling high-quality end-to-end learning of both 3D appearance [25–27] and open-set semantic information [28–31]. However, NeRF-based methods face inherent limitations, including slow training times and difficulties in precise 3D region identification due to their implicit representations. 3DGS [13], an explicit representation, offers efficient and accurate region identification while maintaining reconstruction quality and boosting speed. Although recent works [32–39] have advanced the field, these SfM-based methods struggle with scalability in large scenes. In response, we introduce an expandable semantic representation to current SLAM works, catering to expansion demands.

**Dense Visual SLAM.** Recent advancements [18, 40–47] have integrated NeRFs into SLAM frameworks, including hierarchical multi-feature grids and uncertainty estimation, yet their implicit representations still incur high computational cost. 3DGS leads to a surge in SLAM research [19, 20, 48–60], improving reconstruction quality and efficiency. However, both NeRF- and 3DGS-based methods mainly enhance RGB mapping, neglecting the critical role of semantic representation in expanding the practical applications of SLAM. To bridge this gap, some works [61–68] incorporate semantic information into dense SLAM. Nevertheless, these methods, utilizing closed-set semantic data, lack the adaptability required for continuous integration into any scene. Some latest works [69, 70] employ a label generation approach, still confined to the label-correspondence paradigm, exhibiting limited capability when handling open-set queries (*e.g.*, '*fruit in plastic bag*'). Our work aims to integrate precise open-set semantics into SLAM systems, thereby significantly improving their ability to understand and adapt to arbitrary scenes, enabling more effective applications such as autonomous driving [71] and vision-language navigation [72–75].

**Semantic Pooling.** Directly embedding high-dimensional semantics into 3D Gaussians incurs prohibitive memory cost, a critical issue overlooked by many works [14, 15, 76–81]. Inspired by earlier efforts [82–84], some studies [16, 17] attempt to mitigate this by employing pre-built hard-indexing codebooks with fixed sizes. However, this approach suffers from two fundamental limitations: *(i)* pre-built codebooks cannot dynamically adapt to incremental scene observations; *(ii)* hard-indexing and fixed-size codebooks fail to generalize to novel or rare objects in scalable scenes [85, 86]. To address these challenges, we propose a learnable, soft-aggregated and expandable semantic feature pool, which enables adaptation to sequential SLAM inputs.

**Contrastive Semantic Learning.** Image-level pre-trained models (*e.g.*, CLIP [11]) fail to ensure consistent semantics across frames. While SfM-based methods average semantics across views, overlooking inconsistencies, SLAM systems, processing data sequentially, are more sensitive to such issue. In 2D self-supervised learning, instance discrimination [87] distinguishes images as negative samples [88, 89], excelling in image classification [90–92]. We adopt pixel-level contrastive learning, contrasting intra-object affinities against inter-object ones [93–100], ensuring single-frame semantic consistency. In 3D, self-supervised learning [101–111] is still developing. GarField [29] applies contrastive supervision for only intra-frame consistency, overlooking the problem of inter-frame inconsistencies. Some other works [80] focus only on closed-set classification, lacking open-set semantic field reconstruction. Video-based methods track objects [112–114] and pixels [115–119] across frame, but they often introduce excessive complexity and computational overhead, unsuitable for SLAM. Our approach leverages 3D scene to establish cross-frame pixel correspondence, ensuring semantics consistency while maintaining efficiency and scalability.

## 3 METHODOLOGY

Given an RGB-D stream, our system enhances 3DGS-SLAM frameworks through an expandable semantic representation. Each Gaussian retrieves semantics via key-based aggregation from a shared feature pool, which can dynamically store new semantics and update existing ones (§3.2). During mapping, we optimize semantic parameters through: (1) Intra-Inter Consistency Objectives enforcing cross-view feature stability, and (2) Semantic Stability Guidance preventing inconsistent semantics from affecting reconstructed semantic fields (§3.3). An overview is in Fig. 2.

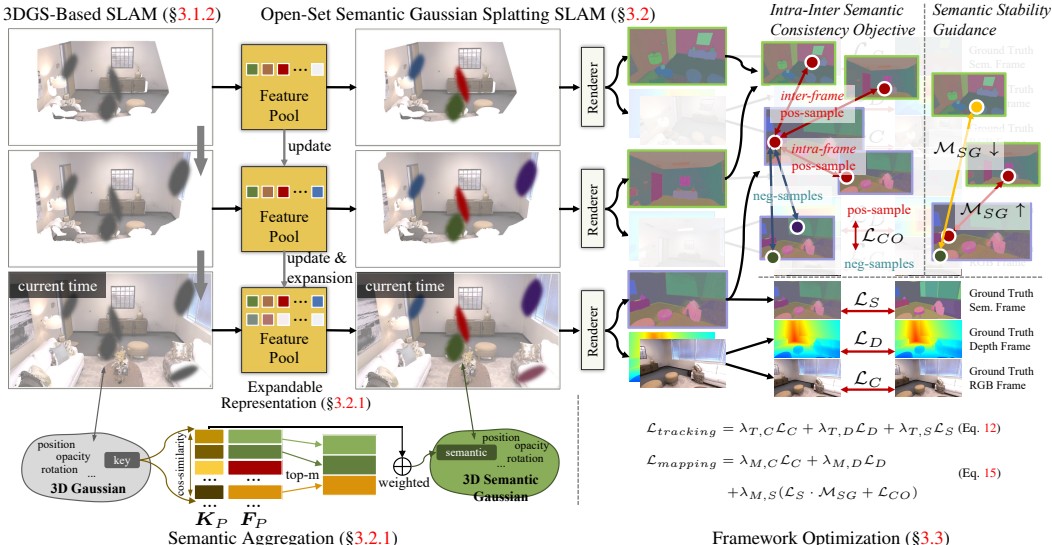

Figure 2: Framework Overview. We enhance existing 3DGS-based SLAM with an expandable semantic representation, introducing a learnable semantic feature pool that stores condensed scene-level semantics and supports dynamic expansion. Each Gaussian retrieves its semantic feature via soft aggregation from the shared pool through a lightweight key. To improve cross-view and temporal consistency, we further introduce an Intra-Inter Semantic Consistency Objective and a Semantic Stability Guidance mechanism, enabling stable and coherent open-set semantic reconstruction during SLAM.

## 3.1 PRELIMINARIES

### 3.1.1 3D GAUSSIAN FIELD

A Gaussian field $\mathcal{G}$ represents an explicit 3D scene with a set of 3D Gaussians:

$$\mathcal{G} := \{\mathcal{G}_i : (\boldsymbol{\mu}_i, \boldsymbol{\Sigma}_i, o_i, \boldsymbol{c}_i)\}_{i=1}^{N}, \tag{1}$$

where each Gaussian $\mathcal{G}_i$ contains its position $\boldsymbol{\mu}_i \in \mathbb{R}^3$, covariance $\boldsymbol{\Sigma}_i \in \mathbb{R}^{3 \times 3}$, opacity $\sigma_i \in [0, 1]$ and color $\boldsymbol{c}_i \in \mathbb{R}^3$. Specifically, given a scaling matrix $\boldsymbol{S}_i \in \mathbb{R}^{3 \times 3}$ and rotation matrix $\boldsymbol{R}_i \in \mathbb{R}^{3 \times 3}$, the covariance matrix $\boldsymbol{\Sigma}_i$ is computed:

$$\boldsymbol{\Sigma}_i = \boldsymbol{R}_i \boldsymbol{S}_i \boldsymbol{S}_i^\top \boldsymbol{R}_i^\top, \tag{2}$$

which describes the shape, size, and orientation of $\mathcal{G}_i$.

**Color Rendering.** Given a viewpoint with its camera pose $T_{CW} \in \mathbb{R}^{4 \times 4}$, the 3D Gaussian $\mathcal{G}_i$ can be projected to 2D plane for rendering [120]:

$$\boldsymbol{\Sigma}_i' = \boldsymbol{J}_i \boldsymbol{W}_i \boldsymbol{\Sigma}_i \boldsymbol{W}_i^\top \boldsymbol{J}_i^\top, \tag{3}$$

where $\boldsymbol{J}_i \in \mathbb{R}^{2 \times 3}$ is the Jacobian matrix of the Gaussian centroid's projection on 2D image plane in camera coordinates, and $\boldsymbol{W}_i \in \mathbb{R}^{3 \times 3}$ is the rotation matrix of $T_{CW}$. The rendered color of a pixel $\boldsymbol{p}$ can be computed via sorting Gaussians by depth and perform front-to-back $\alpha$-blending rendering:

$$\boldsymbol{C}(\boldsymbol{p}) = \sum_{j=1}^{N} \boldsymbol{c}_j \alpha_j \prod_{k=1}^{j-1} (1 - \alpha_k), \tag{4}$$

where $\boldsymbol{c}_j \in \mathbb{R}^3$ is the color of $j$-th Gaussian of the sorted Gaussians, $\alpha_j$ is computed via:

$$\alpha_j = o_j \exp(-\frac{1}{2} \boldsymbol{\sigma}_j^\top (\boldsymbol{\Sigma}_j')^{-1} \boldsymbol{\sigma}_j), \tag{5}$$

where $o_j, \boldsymbol{\sigma}_j$ are the opacity and the offset between $\boldsymbol{p}$ and the center of 2D projection of $j$-th Gaussian.

### 3.1.2 3DGS-BASED SLAM

Given a Gaussian field constructed from frames at time 1 to $t - 1$, along with new RGB and depth frames at $t$, a 3DGS-based SLAM framework performs tracking and mapping.

**Tracking** determines the current camera position with incoming data, optimizing only camera parameters with Gaussian parameters fixed. For each timestep, the camera pose is estimated by forward-projecting pose parameters from the camera center into quaternion space.

**Mapping** generates a scene representation with 3D Gaussians. With the estimated position and the input depth frame, it adds Gaussians to underdeveloped areas, with the poses fixed while Gaussian parameters updated. Large or low-opacity Gaussians are removed.

## 3.2 OPEN-SET SEMANTIC GAUSSIAN SPLATTING SLAM

Our framework extends existing 3DGS-based SLAM with an expandable semantic representation. In §3.2.1, we detail the representation; In §3.2.2, we introduce the rendering process of our framework.

### 3.2.1 EXPANDABLE SEMANTIC REPRESENTATION

We enhance each Gaussian with a semantic feature. To lower memory costs while better adapting to requirements of SLAM, we propose a randomly initialized, learnable and expandable semantic pool.

**Initialization.** Each Gaussian $\mathcal{G}_{Si}$ from a semantic Gaussian field $\mathcal{G}_S$ is assigned a $D_k$-dimensional key $\boldsymbol{k}_i \in \mathbb{R}^{D_k}$. $\mathcal{G}_S$ also contains a $D_k$-dimensional key pool $\boldsymbol{K}_P \in \mathbb{R}^{L \times D_k}$ and a $D_s$-dimensional semantic feature pool $\boldsymbol{F}_P \in \mathbb{R}^{L \times D_s}$, both learnable, of the same size $L$ and $D_k < D_s$:

$$\mathcal{G}_S := \{\mathcal{G}_{Si} : (\boldsymbol{\mu}_i, \boldsymbol{\Sigma}_i, o_i, \boldsymbol{c}_i, \boldsymbol{k}_i), \boldsymbol{K}_P, \boldsymbol{F}_P\}_{i=1}^N. \tag{6}$$

**Aggregation.** To obtain a semantic feature, $\mathcal{G}_{Si}$ first identifies $m$ nearest neighbors (*supp.* §B.8) of $\boldsymbol{k}_i$ from $\boldsymbol{K}_P$ with cosine similarity:

$$\text{NN}_m(\boldsymbol{k}_i, \boldsymbol{K}_P) = \arg\max{}_m(\cos(\boldsymbol{k}_i, \boldsymbol{K}_P)), \tag{7}$$

where $\cos$ and $\arg\max_m$ operate along the size dimension ($L$), and $\text{NN}_m(\boldsymbol{k}_i, \boldsymbol{K}_P) \in \mathbb{R}^m$. We determine importance based on the magnitude of similarity and convert it into weights:

$$\boldsymbol{w}_i = \text{softmax}(\text{NN}_m(\boldsymbol{k}_i \cdot \boldsymbol{K}_P)). \tag{8}$$

Subsequently, $\mathcal{G}_{Si}$ aggregates a semantic feature from $\boldsymbol{F}_P$ with $\boldsymbol{w}_i \in \mathbb{R}^m$:

$$\boldsymbol{s}_i = \boldsymbol{w}_i \cdot \boldsymbol{F}_P', \tag{9}$$

where $\boldsymbol{s}_i \in \mathbb{R}^{D_s}$ represents the semantic feature of $\mathcal{G}_{Si}$, and $\boldsymbol{F}_P' \in \mathbb{R}^{m \times D_s}$ contains $m$ semantics from $\boldsymbol{F}_P$, at the same positions of these $m$ nearest neighbors in $\boldsymbol{K}_P$.

**Expansion.** At initialization, the semantic feature pool $\boldsymbol{F}_P$ and key pool $\boldsymbol{K}_P$ are randomly initialized, and we additionally store a copy of the initial semantic pool, denoted as $\boldsymbol{F}_P^{(0)}$, which serves as a reference for tracking feature utilization. When a new semantic frame is received, we extract unique semantic features and determine whether each of them corresponds to an existing semantic concept. Specifically, we compute cosine similarities between the incoming feature and all entries in the current pool $\boldsymbol{F}_P$. If the top-$m$ similarity scores exceed a predefined threshold, the feature is considered non-novel and will be aggregated by existing pool entries; otherwise, it is marked as novel and scheduled for insertion into the pool. Before inserting novel features, we check whether sufficient capacity exists in $\boldsymbol{F}_P$. Rather than maintaining an explicit usage counter, we adopt a similarity-based criterion to identify *empty slots*. Concretely, for each entry in the current pool $\boldsymbol{F}_P$, we measure its similarity to the corresponding entry in the initial pool $\boldsymbol{F}_P^{(0)}$. Entries whose similarity falls below a predefined threshold are considered to have deviated substantially from their initial random state, indicating that they have accumulated meaningful semantic information and are thus *occupied*. Conversely, entries that remain close to their initial state are treated as empty slots available for storing new semantic features. If the number of available empty slots is insufficient to accommodate the incoming novel features, we expand both the semantic feature pool $\boldsymbol{F}_P$ and the key pool $\boldsymbol{K}_P$ by a factor of $n$. Specifically, we append randomly initialized and learnable segments of size $((n-1)L, D_s)$ and $((n-1)L, D_k)$ to $\boldsymbol{F}_P$ and $\boldsymbol{K}_P$, respectively, resulting in expanded pools $\boldsymbol{F}_P \in \mathbb{R}^{nL \times D_s}$ and $\boldsymbol{K}_P \in \mathbb{R}^{nL \times D_k}$. The corresponding expansion segment is also appended to $\boldsymbol{F}_P^{(0)}$ to maintain a consistent reference for future empty-slot detection. This expansion operation is infrequent in practice and exhibits a self-limiting behavior as the pool gradually captures the semantic diversity of the scene. Implementation details and threshold selection are provided in *supp.* §B.8, and the pseudocode is in Algorithm 1 in APPENDIX.

### 3.2.2 METRIC AND SEMANTIC RENDERING

To render per-pixel color, we employ Eq. 4. Additionally, to optimize the positions of Gaussians, we render the depth of pixel $\boldsymbol{p}$ with a formulation analogous to Eq. 4:

$$\boldsymbol{D}(\boldsymbol{p}) = \sum{}_{j=1}^N \boldsymbol{d}_j \alpha_j \prod{}_{k=1}^{j-1}(1 - \alpha_k), \tag{10}$$

where $\boldsymbol{d}_j$ represents the depth of the $j$-th Gaussian.

To optimize the semantic field with $\boldsymbol{S}_{GT}$, we render a semantic frame after we assign semantics to Gaussians. In contrast, previous codebook-based methods [16, 17] render feature maps first and then assign semantics to each pixel, which means they do not truly construct a 3D semantic field. During rendering, we only select Gaussians within the current view to assign semantics and render the semantic feature of the pixel $\boldsymbol{p}$:

$$\boldsymbol{S}(\boldsymbol{p}) = \sum\nolimits_{j=1}^{N} \boldsymbol{s}_j \alpha_j \prod\nolimits_{k=1}^{j-1} (1 - \alpha_k), \tag{11}$$

where $\boldsymbol{s}_j$ is the allocated semantic feature of the $j$-th Gaussian.

### 3.3 FRAMEWORK OPTIMIZATION

Our framework first distills semantic features with pre-trained models from RGB images (*supp.* §A.1). Given a field reconstructed from time 1 to $t-1$, with new RGB, depth, semantic feature frames $\boldsymbol{C}_{GT}$, $\boldsymbol{D}_{GT}$, $\boldsymbol{S}_{GT}$ and rendered frames $\boldsymbol{C}, \boldsymbol{D}, \boldsymbol{S}$ at $t$, our framework optimizes camera poses via a tracking loss (§3.3.1) and optimize $\boldsymbol{K}_P$, $\boldsymbol{F}_P$ and Gaussian parameters with a mapping loss (§3.3.2).

#### 3.3.1 TRACKING LOSS

For each pixel $\boldsymbol{p}$, camera poses are optimized with:

$$\mathcal{L}_{tracking} = \sum\nolimits_{\boldsymbol{p}} (\lambda_{T,C} \mathcal{L}_{T,C} + \lambda_{T,D} \mathcal{L}_{T,D} + \lambda_{T,S} \mathcal{L}_{T,S}), \tag{12}$$

where $\mathcal{L}_{T,C}$, $\mathcal{L}_{T,D}$, $\mathcal{L}_{T,S}$ are tracking losses for color, depth and semantics, respectively. $\mathcal{L}_{T,S} = \|\boldsymbol{S}(\boldsymbol{p}) - \boldsymbol{S}_{GT}(\boldsymbol{p})\|$, and $\mathcal{L}_{T,C}$, $\mathcal{L}_{T,D}$ are identical to those in the SLAM baseline. For example, in SplaTAM [20], $\mathcal{L}_{T,C} = \|\boldsymbol{C}(\boldsymbol{p}) - \boldsymbol{C}_{GT}(\boldsymbol{p})\|$, $\mathcal{L}_{T,D} = \|\boldsymbol{D}(\boldsymbol{p}) - \boldsymbol{D}_{GT}(\boldsymbol{p})\|$.

#### 3.3.2 MAPPING LOSS

**Intra-Inter Semantic Consistency Objective.** The semantic field is reconstructed from a temporally-ordered sequence of images. However, the semantics are retrieved from 2D open-set models, which often produce viewpoint-dependent and temporally inconsistent predictions. In particular, the same physical object may be assigned different semantic embeddings across frames, since 2D open-vocabulary features are not explicitly aligned over time. This inconsistency leads to contradictory semantic signals in the reconstructed map and hinders the effective learning of a coherent 3D semantic field. To address it, we enforce semantic consistency by progressively aggregating semantic evidence from multiple observations, suppressing noisy or inconsistent predictions across viewpoints.

For a pixel $\boldsymbol{p}$, it should share the same semantic as other pixels from the same object in the current frame (*intra*-frame, $\boldsymbol{S}$), and pixels from the same 3D locations in previous frames (*inter*-frames, $\boldsymbol{S}_r$). Therefore, we select $Q$ neighboring pixels $\{\boldsymbol{p}_i^+\}_{i=1}^Q$ of $\boldsymbol{p}$ from the same object in $\boldsymbol{S}$ and treat their semantics as *intra* positive samples. We then project $\boldsymbol{p}$ into 3D space using depth and reproject it onto $\boldsymbol{S}_r$ to find corresponding pixels $\{\boldsymbol{p}_{i,r}^+\}_{i=1}^R$, whose semantics serve as *inter* positive samples, and $R$ represents the number of selected samples. Semantics of remaining pixels from $\boldsymbol{S}$ are treated as negative samples and added to a negative sample pool $\boldsymbol{N}^-$.

For an *intra* positive sample $\boldsymbol{S}(\boldsymbol{p}_i^+)$, we enhance the similarity between $\boldsymbol{S}(\boldsymbol{p})$ and it, and reduce the similarity between $\boldsymbol{S}(\boldsymbol{p})$ and negative samples $\boldsymbol{S}(\boldsymbol{p}^-)$:

$$\mathcal{L}(\boldsymbol{p}_i^+) = -\log \frac{\exp(\boldsymbol{S}(\boldsymbol{p}) \cdot \boldsymbol{S}(\boldsymbol{p}_i^+))}{\sum_{\boldsymbol{S}(\boldsymbol{p}^-) \in \boldsymbol{N}^-} \exp(\boldsymbol{S}(\boldsymbol{p}) \cdot \boldsymbol{S}(\boldsymbol{p}^-))}. \tag{13}$$

A similar $\mathcal{L}_r(\boldsymbol{p}_{i,r}^+)$ can be calculated for *inter* positive samples $\boldsymbol{S}_r(\boldsymbol{p}_{i,r}^+)$ likewise. Thus, an Intra-Inter Semantic Consistency Objective is computed:

$$\mathcal{L}_{CO} = \frac{1 - \lambda_r}{Q} \sum\nolimits_{i=1}^Q \mathcal{L}(\boldsymbol{p}_i^+) + \frac{\lambda_r}{R} \sum\nolimits_{i=1}^R \mathcal{L}_r(\boldsymbol{p}_{i,r}^+), \tag{14}$$

where $\lambda_r$ is a hyper-parameter to balance intra and inter objectives.

**Semantic Stability Guidance.** To further mitigate the adverse effects of inconsistencies between $\boldsymbol{S}_{GT}$ and the current semantic field, we introduce a Semantic Stability Guidance $\mathcal{M}_{SG}$.

For a pixel $\boldsymbol{p}$ in $\boldsymbol{S}$, we use the same projection rules from the Intra-Inter Semantic Consistency Objective to locate corresponding pixels in previous frames chosen based on the overlap with the current frame, identifying features of the corresponding segmentation regions that corresponding pixels belong to.

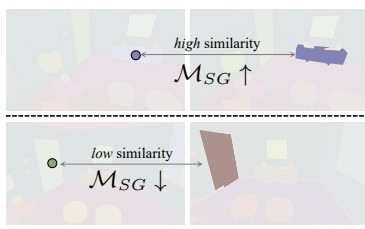

Figure 3: Semantic Stability Guidance.

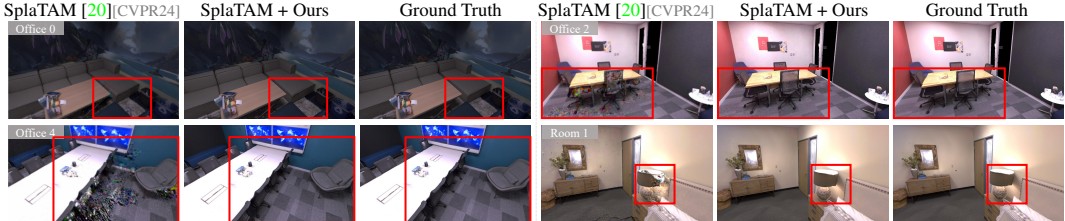

Figure 4: **Qualitative Rendering Quality Comparisons** (§4.2) over SplaTAM [20] on Replica [21].

If $p$ belongs to an object that appears for the first time with no information to guide it, $\mathcal{M}_{SG}(p)$ is set to 1. Otherwise, we calculate the cosine similarity between $S(p)$ and the average feature of all corresponding regions, using this similarity as $\mathcal{M}_{SG}(p)$.

**Overall Mapping Loss.** We jointly optimize the color, depth and semantic feature of $p$:

$$\mathcal{L}_{mapping} = \lambda_{M,C}\mathcal{L}_{M,C} + \lambda_{M,D}\mathcal{L}_{M,D} + \lambda_{M,S}\mathcal{L}_{M,S}, \tag{15}$$

where $\mathcal{L}_{M,C}, \mathcal{L}_{M,D}, \mathcal{L}_{M,S}$ are mapping losses for color, depth and semantics, respectively. $\mathcal{L}_{M,S} = \lambda_S \|S(p) - S_{GT}(p)\| \cdot \mathcal{M}_{SG}(p) + (1 - \lambda_S)\mathcal{L}_{CO}$, and similarly, $\mathcal{L}_{M,C}, \mathcal{L}_{M,D}$ are identical to the SLAM baseline, *e.g.*, in SplaTAM [20], $\mathcal{L}_{M,C} = (1 - \lambda_C) \|C(p) - C_{GT}(p)\| + \lambda_C(1 - \text{SSIM}(C(p), C_{GT}(p)))$, $\mathcal{L}_{M,D} = \|D(p) - D_{GT}(p)\|$.

# 4 EXPERIMENT

## 4.1 EXPERIMENTAL SETUP

**Datasets.** Our evaluation protocol strictly follows the prevailing practice (*e.g.*, SGS-SLAM [64]). Experiments are carried out on eight scenes of Replica [21]. Results on real-world data TUM [121] and ScanNet [122] are delivered in *supp.* §B.

**Implementation Details.** Our experiments run on a server with a single NVIDIA GeForce RTX 3090 GPU. $D_k, D_s, L, Q, R, m, n$ are set to 3, 32, 25, 1, 1, 5, 1 respectively. The optimization involves nine parameters, $\lambda_r = 0.5$, $\lambda_{T,C} = \lambda_{M,C} = 0.5$, $\lambda_{T,D} = \lambda_{M,D} = 1.0$, $\lambda_{T,S} = \lambda_{M,S} = 0.05$, $\lambda_C = 0.2$, and $\lambda_S = 0.999$.

## 4.2 TRACKING AND APPEARANCE MAPPING RESULTS

**Baselines.** We apply our method on SplaTAM [20] and LoopSplat [19], and compare them with Point-SLAM [18], GS$^3$LAM [67], SplaTAM [20], and LoopSplat [19].

**Metrics.** We adopt a standardized set of metrics to evaluate both appearance mapping and camera pose estimation. RGB rendering performance is measured by PSNR, SSIM [123] and LPIPS [124]. Camera pose tracking is assessed by the average absolute trajectory error (ATE RMSE) [121]. Geomatry reconstruction is evaluated using Depth L1.

**Rendering.** Our method outperforms NeRF-based Point-SLAM [18], shown in Table 1a. Compared to 3DGS-based SplaTAM [20] and LoopSplat [19], the inclusion of semantics helps to better capture scene features such as object shapes, further enhancing reconstruction quality. We also surpass current SOTA closed-set semantic SLAM system GS$^3$LAM [67].

**Camera Pose Estimation.** Shown in table 1b, we outperform Point-SLAM [18]; Semantics provides frameworks with more details about the scene, leading to more precise trajectory estimation than SplaTAM [20] and LoopSplat [19].

**Geometric Reconstruction.** As shown in Table 1c, our approach outperforms Point-SLAM [18]. We also surpass SplaTAM [20] and LoopSplat [19], showing that our semantic reconstruction helps to better capture geometry information. Noteworthy, GS$^3$LAM [67] did not report this result in its paper; the data comes from our reproduction of the code. Although GS$^3$LAM [67] introduces closed-set semantic reconstruction, our consistency objective and stability guidance actually enhance the system's sensitivity to geometry, allowing us to significantly outperform it.

**Qualitative Comparisons.** Fig. 4 demonstrate the superior performance of our method. Compared against SplaTAM [20], the integeration of semantics helps to capture scene details such as geometry.

| Methods | Metrics | Avg. | R0 | R1 | R2 | Of0 | Of1 | Of2 | Of3 | Of4 |
|---|---|---|---|---|---|---|---|---|---|---|
| Point-SLAM [18] [ICCV23] | PSNR↑ | 35.17 | 32.40 | 34.08 | 35.50 | 38.26 | 39.16 | 33.99 | 33.48 | 33.49 |
| | SSIM↑ | 0.975 | 0.974 | 0.977 | 0.982 | 0.983 | 0.986 | 0.960 | 0.960 | 0.979 |
| | LPIPS↓ | 0.124 | 0.113 | 0.116 | 0.111 | 0.100 | 0.118 | 0.156 | 0.132 | 0.142 |
| GS³LAM [67] [ACMMM24] | PSNR↑ | 36.26 | 33.67 | 35.80 | 35.96 | 40.28 | 41.21 | 34.30 | 34.27 | 34.59 |
| | SSIM↑ | 0.989 | 0.986 | 0.989 | 0.990 | 0.993 | 0.994 | 0.988 | 0.990 | 0.983 |
| | LPIPS↓ | 0.052 | 0.051 | 0.039 | 0.046 | 0.040 | 0.030 | 0.065 | 0.061 | 0.081 |
| SplaTAM [20] [CVPR24] | PSNR↑ | 34.11 | 32.86 | 33.89 | 35.25 | 38.26 | 39.17 | 31.97 | 29.70 | 31.81 |
| | SSIM↑ | 0.970 | 0.980 | 0.970 | 0.980 | 0.980 | 0.980 | 0.970 | 0.950 | 0.950 |
| | LPIPS↓ | 0.100 | 0.070 | 0.100 | 0.080 | 0.090 | 0.090 | 0.100 | 0.120 | 0.150 |
| **SplaTAM + Ours** | PSNR↑ | 37.61 +3.50 | 34.86 | 37.96 | 37.37 | 41.46 | 42.10 | 35.70 | 35.59 | 35.84 |
| | SSIM↑ | 0.990 +0.020 | 0.985 | 0.990 | 0.990 | 0.992 | 0.993 | 0.990 | 0.990 | 0.986 |
| | LPIPS↓ | 0.050 -0.050 | 0.048 | 0.041 | 0.050 | 0.045 | 0.041 | 0.052 | 0.047 | 0.076 |
| LoopSplat [19] [3DV25] | PSNR↑ | 36.63 | 33.07 | 35.32 | 36.16 | 40.82 | 40.21 | 34.67 | 35.67 | 37.10 |
| | SSIM↑ | 0.985 | 0.973 | 0.978 | 0.985 | 0.992 | 0.990 | 0.985 | 0.990 | 0.989 |
| | LPIPS↓ | 0.112 | 0.116 | 0.122 | 0.111 | 0.085 | 0.123 | 0.140 | 0.096 | 0.106 |
| **LoopSplat + Ours** | PSNR↑ | 37.95 +1.32 | 34.95 | 38.02 | 37.54 | 42.01 | 42.20 | 35.84 | 35.89 | 37.12 |
| | SSIM↑ | 0.989 +0.004 | 0.979 | 0.987 | 0.990 | 0.993 | 0.993 | 0.989 | 0.993 | 0.989 |
| | LPIPS↓ | 0.053 -0.059 | 0.056 | 0.048 | 0.060 | 0.047 | 0.045 | 0.060 | 0.057 | 0.057 |

(a) Rendering Performance (§4.2).

| Methods | Avg. | R0 | R1 | R2 | Of0 | Of1 | Of2 | Of3 | Of4 |
|---|---|---|---|---|---|---|---|---|---|
| Point-SLAM [18] | 0.52 | 0.61 | 0.41 | 0.37 | 0.38 | 0.48 | 0.54 | 0.69 | 0.72 |
| GS³LAM [67] | 0.37 | 0.27 | 0.25 | 0.28 | 0.67 | 0.21 | 0.33 | 0.30 | 0.65 |
| SplaTAM [20] | 0.36 | 0.31 | 0.40 | 0.29 | 0.47 | 0.27 | 0.29 | 0.32 | 0.55 |
| **SplaTAM + Ours** | 0.29 -0.07 | 0.24 | 0.29 | 0.28 | 0.29 | 0.20 | 0.25 | 0.30 | 0.46 |
| LoopSplat * [19] | 0.26 | 0.28 | 0.22 | 0.17 | 0.22 | 0.16 | 0.49 | 0.20 | 0.30 |
| **LoopSplat + Ours** | 0.23 -0.03 | 0.25 | 0.20 | 0.16 | 0.21 | 0.16 | 0.34 | 0.19 | 0.29 |

*The method implements loop closure and global bundle adjustment.

(b) Camera Pose Estimation (§4.2) (ATE RMSE↓ [cm]).

| Methods | Avg. | R0 | R1 | R2 | Of0 | Of1 | Of2 | Of3 | Of4 |
|---|---|---|---|---|---|---|---|---|---|
| Point-SLAM [18] | 0.44 | 0.53 | 0.22 | 0.46 | 0.30 | 0.57 | 0.49 | 0.51 | 0.46 |
| GS³LAM [67] | 0.52 | 0.64 | 0.44 | 0.59 | 0.47 | 0.36 | 0.48 | 0.50 | 0.65 |
| SplaTAM [20] | 0.72 | 0.43 | 0.38 | 0.54 | 0.44 | 0.66 | 1.05 | 1.60 | 0.68 |
| **SplaTAM + Ours** | 0.34 -0.38 | 0.42 | 0.26 | 0.38 | 0.26 | 0.15 | 0.32 | 0.44 | 0.49 |
| LoopSplat [19] | 0.51 | 0.39 | 0.23 | 0.52 | 0.32 | 0.51 | 0.63 | 1.09 | 0.40 |
| **LoopSplat + Ours** | 0.30 -0.21 | 0.34 | 0.21 | 0.35 | 0.24 | 0.27 | 0.30 | 0.35 | 0.36 |

(c) Reconstruction Performance (§4.2) (Depth L1↓ [cm]).

Table 1: **Quantitative Tracking and Appearance Mapping Results** on Replica [21].

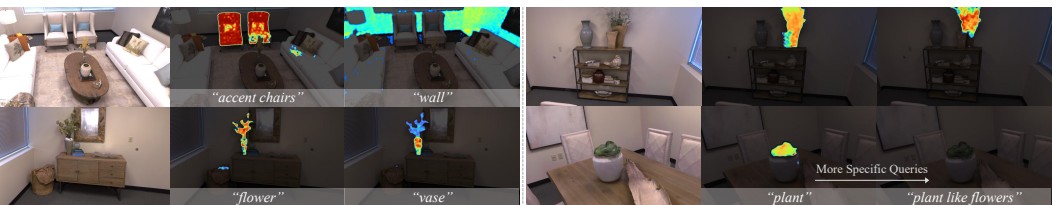

Figure 5: **Qualitative Results of Object Localization** (§4.4) on Replica [21]. ("accent chairs", "wall", "flower", "vase", More Specific Queries, "plant", "plant like flowers")

## 4.3 CLOSED-SET SEMANTIC MAPPING RESULTS

**Baselines.** We compare our method with SNI-SLAM [63], SGS-SLAM [64], GS³LAM [67] and Hier-SLAM++ [68], using the basic framework of SplaTAM [20].

**Metrics.** We employ the mean Intersection over Union (mIoU) as our evaluation metric.

**Results.** Table 2 presents comparisons across four scenes from Replica [21]. For fair comparisons, our results are obtained with a classification head on top of semantic features (*supp.* §A.2), supervised by ground truth labels. Our method outperforms NeRF-based SNI-SLAM [63]. Furthermore, by integrating the Intra-Inter Semantic Consistency Objective and Semantic Stability Guidance, our framework maintains consistent semantics across frames, and surpasses 3DGS-based frameworks. We also conduct experiments on open-set semantic segmentation (§4.4), which is not possible with previous closed-set SLAM.

| Methods | Closed-set | | | | | Open-set |
|---|---|---|---|---|---|---|
| | Avg. | R0 | R1 | R2 | Of0 | 2D mIoU |
| SNI-SLAM [63] | 87.41 | 88.42 | 87.43 | 86.16 | 87.63 | N/A |
| SGS-SLAM [64] | 92.72 | 92.95 | 92.91 | 92.10 | 92.90 | |
| GS³LAM [67] | 96.63 | 96.83 | 96.68 | 96.40 | 96.61 | |
| Hier-SLAM++ [68] | 90.35 | 91.21 | 90.62 | 89.11 | 90.45 | |
| **SplaTAM + Ours** | 96.76 | 96.93 | 96.82 | 96.67 | 96.62 | 66.5 |

Table 2: **Comparisons with semantic SLAM** (§4.3).

## 4.4 OPEN-SET SEMANTIC MAPPING RESULTS

**Datasets.** Experiments are carried out on Replica [21] and in-the-wild data captured by our everyday devices. Details on our in-the-wild data are presented in *supp.* §B.6.

**Baselines.** We apply our method based on SplaTAM [20]. Our competitors include SfM-based methods (*i.e.*, LERF [28], LEGaussians [16] and GOI [17]).

**Metrics.** For segmentation, we employ mIoU as our metric, same as in GOI [17]. For localization, we use acc@0.25 and acc@0.5 as metrics. Details on localization evaluations are in *supp.* §B.7.

**Semantic Segmentation & Localization.** Comparisons with SfM-based methods are in Table 3. While the baselines require COLMAP and full image sequences for camera pose estimation, we estimate poses online solely from RGB-D input. Results demonstrate that we construct a more accurate semantic field. We also conduct experiments on 3D localization, where we match the similarity between input text and Gaussians to identify the most relevant Gaussian sets. Other codebook-based methods [16, 17] cannot construct an explicit 3D semantic field and interact with Gaussians directly, consistent with our previous discussions (§3.2.2).

| Methods | Replica [21] | | | In-the-wild data | | |
|---|---|---|---|---|---|---|
| | 2D mIoU | 3D Localization acc@0.25 | acc@0.5 | 2D mIoU | 3D Localization acc@0.25 | acc@0.5 |
| LERF [28] | 28.2 | N/A | | 20.6 | N/A | |
| LEGaussians [16] | 39.4 | | | 32.1 | | |
| GOI [17] | 61.7 | | | 49.5 | | |
| **SplaTAM + Ours** | 66.5 | 54.7 | 45.1 | 60.3 | 51.2 | 42.9 |

Table 3: **Comparisons with SfM-based Open-Set Semantic Methods** (§4.4).

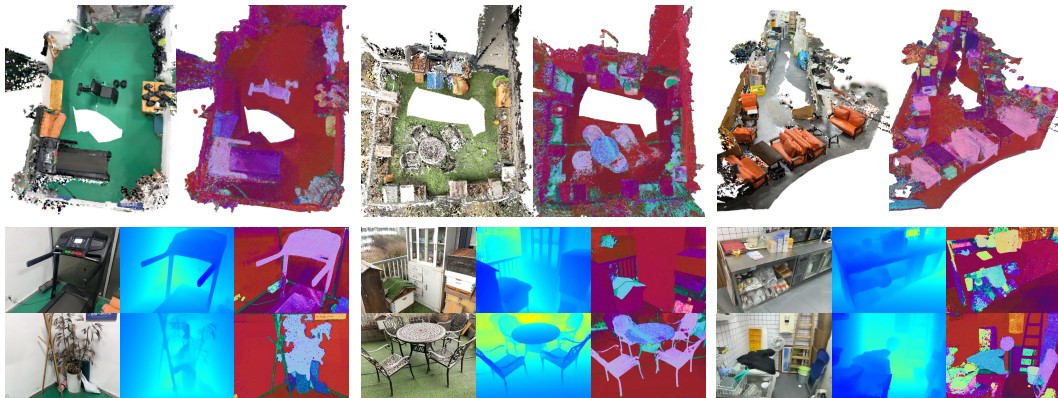

Figure 8: **Visualizations** (§4.4) on three in-the-wild scenes captured by everyday devices. *Top*: Whole scenes. *Bottom*: Rendered images. Details on the data of scenes are in *supp.* §B.6.

**Open-set Semantic Visualizations.** Visualizations of RGB, depth, open-set and closed-set semantics on Replica [21] *Room 0* are in Fig. 6. As seen, we are able to reconstruct high-quality appearnce with high geometric accuracy. We are also able to reconstruct an open-set semantic field with clear edges, which is beneficial for downstream tasks such as 3D scene editing.

**Open-Vocabulary Object Localization.** Given an open-vocabulary query, our method generates highly concentrated activation regions that align precisely with the ground truth shape. Moreover, as shown in Fig. 5, our field can identify specific

Figure 6: **Visualizations** (§4.4) of RGB, depth, open-set and closed-set semantics on Replica [21] *Room 0.*

objects when provided with both general queries (*e.g.*, *"plant"*) and specific queries (*e.g.*, *"plant like flowers"*), a capability that closed-set methods lack.

**3D Editing.** We evaluate our open-set semantic representation for 3D editing in Fig. 7. Our method achieves object editing while maintaining the integrity of the surrounding environment, showing the accuracy of our semantic field.

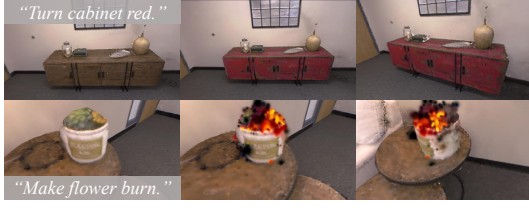

**Everyday Devices Results.** We evaluate our framework on four in-the-wild scenes captured by our hand-held everyday devices in Fig. 1 and

Figure 7: **Results of 3D Editing** (§4.4) on Replica [21].

8, demonstrating the ability of constructing dense color maps and open-set semantic fields with clear boundaries in noval scenes with only RGB-D data, even without any additional annotation.

## 4.5 DIAGNOSTIC EXPERIMENTS

Experiments are conducted on Replica *Room 0* based on SplaTAM.

**Key Component Analysis.** In Table 4, we validate the importance of our proposed components by attaching them one at a time. The 1st row directly assigns a 3-dim semantic to each Gaussian. In the 2nd row, we utilize $\boldsymbol{F}_p$, resulting in performance improvements. The 3rd row and the 4th row attach $\mathcal{L}_{CO}$ and $\mathcal{M}_{SG}$ respectively, further improving performance, which shows that our method can effectively ensure semantic consistency.

|  | PSNR ↑ | ATE RMSE ↓ | mIoU ↑ |
|---|---|---|---|
| *w/o* All | 32.88 | 0.30 | 72.56 |
| $+ \boldsymbol{F}_P$ | 33.40 | 0.26 | 80.15 |
| $+ \mathcal{L}_{CO}$ | 33.67 | 0.25 | 84.23 |
| $+ \mathcal{M}_{SG}$ | **33.81** | **0.24** | **87.29** |

Table 4: Key Component Analysis ($D_s$=3) (§4.5).

**Semantic Feature Dimension.** We examine how the semantic feature dimension affects semantic reconstruction in Table 5, where X = 16.0 GB. Without $\boldsymbol{F}_P$, when $D_s$ = 6, allocating a feature to each Gaussian incurs substantial memory consumption, leads to experimental failure. While with $\boldsymbol{F}_P$, we are able to utilize features with larger

|  | $D_s$ | GPU Mem. Usage ↓ | PSNR↑ | mIoU↑ |
|---|---|---|---|---|
| *w/o* $\boldsymbol{F}_P$ | 3 | X GB | 32.88 | 72.56 |
|  | 6 | ∼(X + 12.0) GB | - | - |
| *w/* $\boldsymbol{F}_P$ | 3 | (X + 1.5) GB | 33.81 | 87.29 |
|  | 8 | (X + 1.7) GB | 33.97 | 88.95 |
|  | **32** | **(X + 2.4) GB** | **34.86** | **96.93** |
|  | 64 | (X + 3.7) GB | 34.67 | 96.73 |

Table 5: Semantic Feature Dimension (§4.5).

dimension than those without $F_P$, *i.e.*, 64 *vs.* 3, showcasing the superior semantic compression capabilities of our pool. Additionally, low-dim features are inadequate for high-quality reconstruction, while higher dimensions provide diminishing returns.

**Semantic Pool Design.** We compare four pool designs: *i)* *implicit* design of sequential internal state updates through LSTM-style gates; *ii)* *fixed-size* explicit pool updated via gating mechanisms similar to LSTM; *iii)* *compressive* explicit pool employing content-aware compression for capacity management; and *iv)* *expandable* explicit pool as our

|  |  | mIoU ↑ | Training Time ↓ | GPU Mem. Usage ↓ |
|---|---|---|---|---|
| Baseline | | 72.56 | X h | Y GB |
| Implicit Method | | 77.29 | (X + 0.6) h | (Y + 0.5) GB |
| Explicit | Fixed-Sized | 86.94 | (X + 1.8) h | (Y + 1.8) GB |
| | Compressive | 88.62 | (X + 2.7) h | (Y + 2.1) GB |
| | Expandable | **96.93** | **(X + 1.9) h** | **(Y + 2.4) GB** |

Table 6: Semantic Pool Design (§4.5).

default setup. Baseline employs no pool with $D_s = 3$, and X = 6.0 h, Y = 16.0 GB. As shown in Table 6, our expandable pool achieves superior accuracy with moderate memory and computation cost, maintaining practical resource utilization. The implicit variant achieves fastest training and minimal memory consumption, but suffers severe accuracy degradation. Fixed-size variant fails to maintain stable performance across varying scenes, and compressive pool loses fine-grained temporal information. Besides, surprisingly, expandable pool achieves faster speed compared to compressive pool, due to avoided compression overhead.

**Large-Scale Scenes Performance.** We conduct experiments on a large-scale in-the-wild scene comprising 12,000 frames (*supp.* §B.6), comparing our method with SGS-SLAM [64] and GS³LAM [67]. Since both SGS-SLAM and GS³LAM require supervision from labels, we use generated pseudo-semantic labels (*supp.* §B.8). Shown in Table 7, both SGS-SLAM and GS³LAM, which assign 32-dim features to each

| Method | Complete | GPU Mem. Usage ↓ |
|---|---|---|
| SGS-SLAM [64] | ✗ | >24 GB |
| GS³LAM [67] | ✗ | >24 GB |
| **SplaTAM + Ours** | ✓ | **19.5 GB** |

Table 7: Large-Scale Scenes Performance (§4.5).

Gaussian, terminate prematurely due to GPU memory overflow. In contrast, our approach runs stably until full reconstruction. During this process, the size of $F_P$ expands from 25 to 400 to accommodate additional semantics. Despite this significant expansion, our method still maintains lower memory consumption compared to direct assignment. These results demonstrate our method's effectiveness in handling large-scale environments and long sequence inputs.

## 5 CONCLUSION

We present Open-Set Semantic Gaussian Splatting SLAM, enabling everyday devices to reconstruct 3D scenes with rich and expandable open-set semantics. By aggregating semantics for 3D Gaussians through a shared and lightweight representation, our method significantly reduces memory costs while supporting continuous and scalable semantic updates during SLAM. Furthermore, the proposed consistency-aware optimization improves cross-view semantic coherence, leading to more stable and reliable semantic fields over time. Experiments demonstrate high-fidelity rendering quality together with scalable semantic reconstruction across diverse scenes. This work advances accessible, high-quality, and expressive 3D world modeling, and provides a practical foundation for semantic-aware applications such as open-vocabulary localization and interactive 3D scene understanding.

## 6 LIMITATIONS

This work focuses on introducing scalable open-set semantic reconstruction into SLAM systems, rather than redesigning the tracking or mapping back-end. As a result, our framework is implemented as a modular extension on top of existing SLAM pipelines, and its runtime performance as well as robustness are largely bounded by the underlying baseline system (see *supp.* §B.4). While this design enables broad compatibility with different 3DGS-based SLAM frameworks, it is also possible that further improvements in efficiency, robustness, or real-time performance would likely require a tighter co-design of the SLAM back-end and the proposed semantic components.

In addition, similar to most existing dense SLAM systems, our approach assumes static or weakly dynamic environments and may exhibit limited robustness in highly dynamic or extremely large-scale settings, such as scenes with frequent object motion or crowded environments. Although the expandable semantic pool and the proposed consistency-aware optimization provide a foundation for aggregating object-level semantics over time, explicit modeling of object motion and long-term semantic drift is beyond the scope of the current system. Addressing these challenges would require additional mechanisms, such as dynamic-aware mapping, motion segmentation, or temporal semantic association, which we leave for future work.

## ACKNOWLEDGEMENT

This work was supported by "Pioneer" and "Leading Goose" R&D Program of Zhejiang (No. 2024C01161), Fundamental Research Funds for the Central Universities (226-2025-00057), CIE-Tencent Robotics XRhino-Bird Focused Research Program, China Postdoctoral Science Foundation (No. 2025T180421), Postdoctoral Fellowship Program of CPSF (No. GZC20251066), and Postdoctoral Science Preferential Funding of Zhejiang (No. ZJ2025001).

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

# APPENDIX

This appendix contains additional details for the ICLR 2026 submission, titled *Open-Set Semantic Gaussian Splatting SLAM with Expandable Representation*. The appendix is organized as follows:

- §A provides more details on our method.
- §B presents more experiment results.
- §C gives our pseudocode and limitations.
- §D offers asset licenses and societal impact.

## A  MORE DETAILS ON METHODS

### A.1  PREPROCESS RGB FRAMES

We generate semantic maps $S_{GT}$ using RGB frames $C_{GT}$. We first generate segment masks with an image segmentation model SAM [12]. We then extract pixel-aligned CLIP [11] features for each segmented object. Notably, our framework is not dependent on any specific semantic extraction method and can alternatively employ other segmentation approaches (*e.g.*, SAM 2 [125]), CLIP models (*e.g.*, EVACLIP [126]), or unified semantic extraction methods (*e.g.*, OVSAM [127]). Therefore, this preprocessing pipeline is an implementation choice for semantic initialization of our framework.

### A.2  FRAMEWORK DETAILS

**Auxiliary Semantic Head.** Given a 2D semantic feature map $S$ rendered from semantic features, we train an semantic head to perform a classification task. Specifically, for each pixel $p$ in $S$, we classify it into one of the annotated categories, resulting in a closed-set semantic map $\hat{S}(p)$:

$$\hat{S}(p) = \text{classifier}(S(p)). \tag{16}$$

Supposed we have the input ground truth closed-set semantic map $\hat{S}_{GT}$ for currrent time $t$, we can optimize the classfier with a cross-entropy loss:

$$\mathcal{L}_{\hat{S}} = \mathcal{L}_{ce}(\hat{S}, \hat{S}_{GT}). \tag{17}$$

We implement the classification for the quantitative comparisons mentioned in §4.3.

**Selected Mapping Frames and Keyframes.** Instead of performing mapping at every frame, we map only at selected frames. Optimization of Gaussians typically prioritizes recent frames, which can cause the system to forget earlier optimization results. To mitigate this, we trigger mapping only when a significant number of new Gaussians need to be added to the scene in the current frame. Additionally, if no mapping occurs for several consecutive frames, we designate the current frame as a mapping frame to ensure full scene optimization.

We also maintain a set of keyframes, storing every *n*-th frame along with the current frame. During mapping, we evaluate the overlap between the current frame and each keyframe using the current depth map, selecting the most relevant keyframes for optimization. One of them is randomly chosen for each iteration of mapping.

### A.3  DETAILS ON 3D SCENE EDITING

Previous research has given limited attention to selective editing of individual objects within complex scenes. Most existing methods [128–133] for such scenes focus primarily on object movement or removal. Although diffusion-based methods [134–138] can alter the appearance or style of objects, they are typically limited to small areas around the target object. These diffusion-based methods face challenges: on one hand, they require the target object to be clearly visible, which complicates editing in scenes with multiple objects, as diffusion may struggle to isolate the intended target (*supp.* §A.3.1). On the other hand, diffusion models often unintentionally modify parts of non-target objects, resulting in unwanted changes in larger, open scenes (*supp.* §A.3.2).

### A.3.1  CAMERA GENERATION

**Camera Positions.** Given a center coordinate $o$, an initial upper vector $v_0$, we can generate a sphere with radius $r$ in spherical coordinates $(r, \theta, \varphi)$. The angles are typically based on $v_0$ as the "north pole". However, for more flexible editing, we often need to specify an additional normal vector $v_1$

as the "north pole" to rotate the sphere. This enables alignment of the generated camera positions with the desired orientation, accommodating scenarios such as objects on walls, where the north pole should be perpendicular to the wall rather than the ground. To transform the sphere, we can obtain the rotation matrix between $v_0$ and $v_1$ with Rodrigues' rotation formula:

$$\boldsymbol{R} = \boldsymbol{I} + \boldsymbol{K} + \boldsymbol{K}^2 \frac{1-c}{s^2}, \tag{18}$$

where

$$\boldsymbol{v} = \boldsymbol{v}_0 \times \boldsymbol{v}_1, \quad c = \boldsymbol{v}_0 \cdot \boldsymbol{v}_1, \quad s = \|\boldsymbol{v}\|,$$

$$\boldsymbol{K} = \begin{bmatrix} 0 & -\boldsymbol{v}_z & \boldsymbol{v}_y \\ \boldsymbol{v}_z & 0 & -\boldsymbol{v}_x \\ -\boldsymbol{v}_y & \boldsymbol{v}_x & 0 \end{bmatrix}. \tag{19}$$

Now we can transform any point to the place we need on the surface of the sphere with $\boldsymbol{R}$. We divide the radian values of the spherical coordinate system $\theta, \varphi$ evenly to generate camera positions on the sphere, and transform them to Cartesian coordinate positions $\boldsymbol{c} = (x, y, z)$:

$$x = r \sin\theta \cos\varphi, \quad y = r \sin\theta \sin\varphi, \quad z = r \cos\theta. \tag{20}$$

**Camera Pose Matrix.** With the position $\boldsymbol{c}$ of a generated camera and the center coordinate $\boldsymbol{o}$, we have the look direction vector $\boldsymbol{l} = \boldsymbol{o} - \boldsymbol{c}$, which indicates the direction the camera is pointing. Given an upper vector (such as $\boldsymbol{v}_0$) $\boldsymbol{u}$, we derive the right vector as $\boldsymbol{r} = \boldsymbol{u} \times \boldsymbol{l}$. To ensure the upper vector is completely orthogonal to both the look direction and right vector, we compute a corrected upper vector $\boldsymbol{u}' = \boldsymbol{l} \times \boldsymbol{r}$, which ensures $\boldsymbol{r}, \boldsymbol{u}', \boldsymbol{l}$ form an orthonormal basis, representing the $x, y, z$ axes of the camera coordinate system respectively. Then we have the view matrix:

$$\boldsymbol{V} = \begin{bmatrix} \boldsymbol{r}_x & \boldsymbol{u}'_x & -\boldsymbol{l}_x & -\boldsymbol{r} \cdot \boldsymbol{c} \\ \boldsymbol{r}_y & \boldsymbol{u}'_y & -\boldsymbol{l}_y & -\boldsymbol{u} \cdot \boldsymbol{c} \\ \boldsymbol{r}_z & \boldsymbol{u}'_z & -\boldsymbol{l}_z & -\boldsymbol{l} \cdot \boldsymbol{c} \\ 0 & 0 & 0 & 1 \end{bmatrix}. \tag{21}$$

After generating a sequence of cameras positioned around the target object, we utilize the Gaussian differentiable renderer to render images centered on the object, meeting the requirements for diffusion-based editing.

### A.3.2 PRECISE EDITING

Editing specific objects without affecting surrounding areas often relies on using *2D* masks to restrict loss computation. However, this approach faces key challenges. **First**, *updating* masks during editing incurs high computational costs and leads to progressively inaccurate mask generation. In contrast, *static* masks may fail to support precise edits, especially for modifications extending beyond the mask's boundaries. **Second**, during optimization, the impact is not confined solely to Gaussians belonging to the target object; parts of the scene occluded by the object but still within the mask region may also be unintentionally altered.

To address these challenges, our methodology leverages the embedded semantics in our representation. We employ the CLIP model [11] to extract features from the input text, evaluate their similarity to the semantics that Gaussians retrieve from the feature pool, and selectively refine the relevant Gaussians for the editing. This approach effectively improves editing precision by directly interacting with the scene's underlying representation.

During the editing process, we randomly select a view obtained from *supp.* §A.3.1 to render the original image $I_{ori}$, and utilize InstructPix2Pix [139] to generate the edited image $I_{edit}$. We then compute the perceptual loss [140] between them to optimize the scene:

$$\mathcal{L}_{editing} = \mathcal{L}_{perceptual}(I_{ori}, I_{edit}). \tag{22}$$

At each iteration, views are re-selected and editing is applied to the previously edited scene. This iterative approach minimizes inconsistencies that can arise from editing all view images simultaneously, allowing us to make further optimizations on the already edited scene.

### A.3.3 REMOVAL AND MOVEMENT

**Removal.** We remove an object using a similar approach to editing. We first extract the object's CLIP feature to identify its semantics and remove the corresponding Gaussians. Then, we render images from the generated camera poses and inpaint them using diffusion. During optimization, to avoid altering the surrounding environment, we only update Gaussians within the sphere centered at $o$ with radius $r$ (*supp.* §A.3.1). At each iteration, we re-select a camera to continuously inpaint the scene.

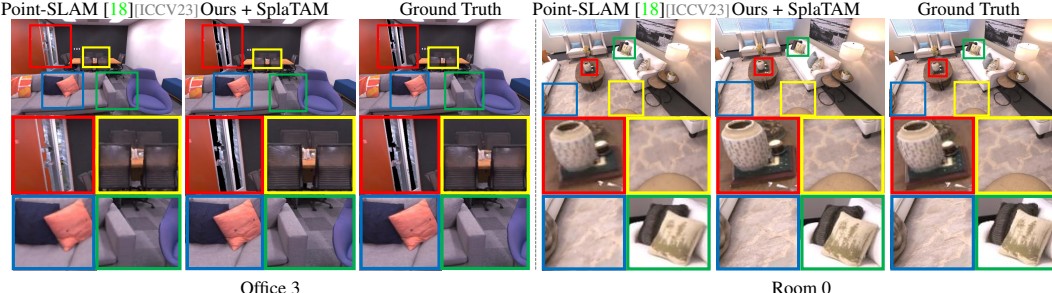

Point-SLAM [18][ICCV23] Ours + SplaTAM    Ground Truth     Point-SLAM [18][ICCV23] Ours + SplaTAM    Ground Truth

Office 3                        Room 0

Figure 9: **Qualitative Comparisons** (§B.2.1) over Point-SLAM [18] on Replica [21].

**Movement.** We break the movement into two stages. First, similar to removal, we locate the target object, select its Gaussians, and perform inpainting at its original position. Second, we move the selected Gaussians by applying the input translation and rotation to their coordinates.

# B MORE EXPERIMENT RESULTS

## B.1 EXPERIMENTAL SETUP

**Datasets.** The experiment results in this part are conducted on Replica [21], TUM-RGBD [121] and ScanNet [122], with evaluations on 8, 5 and 6 scenes.

**Implementation Details** are the same in §4.1. To be more specific, during tracking, only camera parameters are optimized, with learning rates of 2e-3 for translations, 4e-4 for unnormalized rotations. During mapping, only Gaussian parameters and $K_P$, $F_P$ are optimized, with learning rates of 9e-5 for 3D positions, 2.5e-3 for colors, 1e-1 for semantic-embedded parameter, 1e-2 for semantic feature pool, 1e-3 for key pool, 8e-4 for rotations, 4e-2 for opacities, and 8e-4 for scales.

## B.2 MORE TRACKING AND APPEARANCE MAPPING RESULTS

**Baselines.** We apply our method on SplaTAM [20], LoopSplat [19], and more recent works S3PO-GS [59] and SEGS-SLAM [60].

**Metrics.** We assess camera tracking and mapping with the same metrics as in §4.2.

### B.2.1 MORE RESULTS ON REPLICA

**More Quantitative Comparisons.** We present more quantitative comparisons with more recent works in Table 8. Notably, S3PO-GS [59] is designed for open-world scenarios, and the Replica results shown here come from our reproduction.

**More Qualitative Comparisons.** We present more visual comparisons in Fig. 4. Compared to Point-SLAM [18], our method excels at capturing finer details, such as patterns, wrinkles, plush textures, and even lighting variations.

| Method | PSNR ↑ | SSIM ↑ | LPIPS ↓ | ATE RMSE ↓ |
|---|---|---|---|---|
| S3PO-GS [59][ICCV25] | 32.71 | 0.943 | 0.138 | 2.26 |
| **S3PO-GS + Ours** | **35.23** | **0.961** | **0.070** | **0.97** |
| SEGS-SLAM [60][ICCV25] | 39.42 | 0.975 | 0.021 | 0.43 |
| **SEGS-SLAM + Ours** | **39.88** | **0.983** | **0.020** | **0.38** |

Table 8: **More Quantitative Comparisons** (*supp.* B.2.1) with more recent works on Replica [21].

**Other Visualizations.** Fig. 13 shows more qualitative results on color, depth and semantic field reconstruction. Our method produces high-quality color field reconstructions on Replica [21], which is also reflected in geometric details. We also reconstruct semantic fields with clear boundaries, enabling more applications like 3D scene editing.

### B.2.2 RESULTS ON SCANNET AND TUM

**Camera Pose Estimation.** As shown in Table 18, our method outperforms existing NeRF-based approaches, even on the challenging TUM, where sparse depth information and severe motion blur are common. Despite these similar challenges, our approach also achieves competitive results on ScanNet when compared to the latest methods.

**Quantitative Analysis of Scene Reconstruction.** The rendering results on real-world data, shown in Table 19, highlight the superior performance of our method compared to previous approaches,

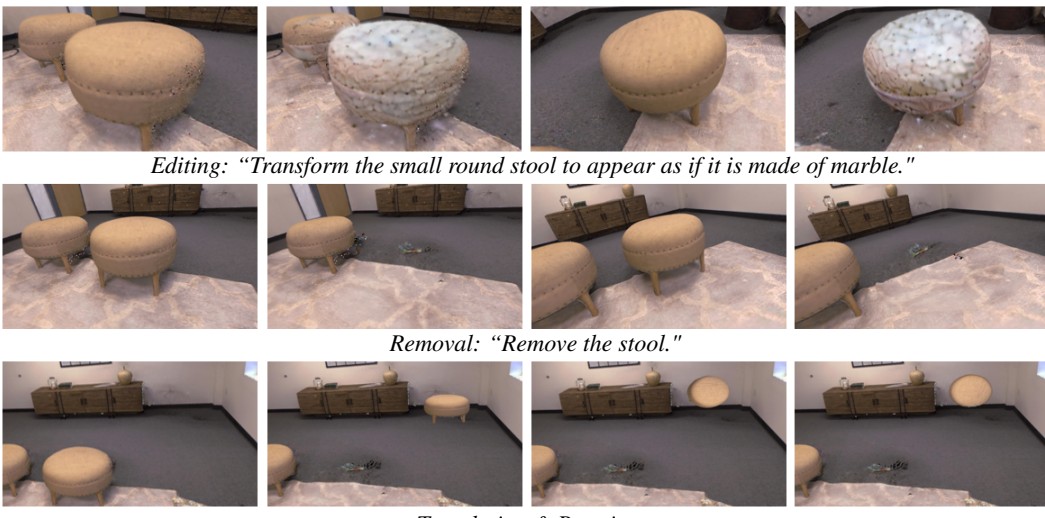

*Editing: "Transform the small round stool to appear as if it is made of marble."*

*Removal: "Remove the stool."*

*Translation & Rotation*

Figure 10: **Qualitative Results of 3D Editing** (*supp.* §B.5) on Replica [21].

underscoring its effectiveness in real-world settings. Due to the lack of semantic annotation in TUM, GS³LAM [67] cannot be run on this dataset.

**More Visualizations.** We present more visual results on ScanNet [122] and TUM-RGBD [121] are shown in Fig. 14. As seen, our method produces high-quality color and semantic field reconstructions on real-world data. Notably, while previous closed-set semantic methods struggle with datasets that lacks semantic annotations such as TUM, our approach successfully reconstructs semantic fields across any scene or dataset, enabling broader application possibilities.

### B.3 MORE CLOSED-SET SEMANTIC MAPPING RESULTS

**Datasets and Baselines.** Following GS³LAM [67], we use real-world ScanNet [122] scene0059_00, which is also designated as an example in their official GitHub repository.

**Metrics.** We employ mIoU as our metric.

**Results.** We use the same claasification head as mentioned in *supp.* §A.2. Table 9 demonstrates that our method outperforms previous work on real-world data, which can be attributed to the introduction of our consistent open semantics.

| Method | mIoU ↑ |
|---|---|
| GS³LAM [67] | 18.62 |
| **SplaTAM + Ours** | **53.27** |

Table 9: **Closed-set comparisons** (*supp.* B.3) with GS³LAM [67] on ScanNet [122] scene0059_00.

### B.4 RUNTIME ANALYSIS

Table 10 presents the runtime of our framework. Due to the introduction of semantic reconstruction, our method incurs approximately 8% – 21% additional time overhead compared to SplaTAM [20], LoopSplat [19] and MonoGS [48]. For tracking, the additional computational overhead stems from semantic assignment from $F_P$; For mapping, it includes semantic assignment, Intra-Inter Semantic Consistency Objective and Semantic Stability Guidance. Notably, the additional overhead introduced

| Method | FPS ↑ | Tracking /Frame (s) ↓ | Mapping /Frame (s) ↓ |
|---|---|---|---|
| SplaTAM | 0.23 | 1.78 | 2.56 |
| **SplaTAM + Ours** | 0.16 | 2.33 | 3.82 |
| LoopSplat | 0.57 | 0.83 | 0.93 |
| **LoopSplat + Ours** | 0.37 | 0.97 | 1.72 |
| MonoGS | 1.08 | 0.36 | 0.57 |
| **MonoGS + Ours** | 0.99 | 0.37 | 0.65 |

Table 10: **Runtime** (*supp.* B.4) on Replica Room 0.

for MonoGS is significantly lower than that for SplaTAM and LoopSplat. This is because MonoGS requires far fewer Gaussians for reconstruction compared to them ($3 \times 10^5$ *v.s.* $5 \times 10^6$), reducing the computational burden of semantic assignment and consistency optimization. This observation also suggests that as more compact 3DGS techniques emerge, our framework's efficiency can be further improved.

### B.5 MORE RESULTS ON 3D SCENE EDITING

Fig. 10 shows more results of 3D scene editing. Beyond the editing mentioned in §4.4, our method also supports removal, translation and rotation, enabled by accurate semantic fields we construct.

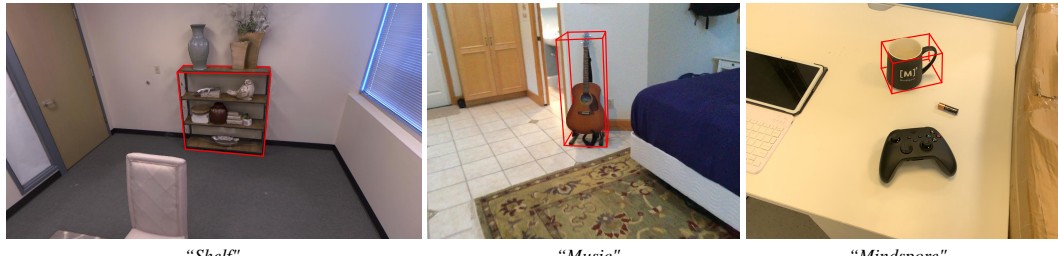

*"Shelf"*        *"Music"*        *"Mindspore"*

Figure 11: **Annotation examples** (*supp.* §B.7) on three datasets.

## B.6 DETAILS ON IN-THE-WILD DATA

We collected data using an iPad equipped with a LiDAR. This dataset includes RGB-D images from 9 scenes, without any extrinsic camera parameters. These scenes are generally more complex than those used in current works, such as scenes in ScanNet [122]. In terms of scale, the number of images per scene ranges from 5,000 to 12,000 frames. Regarding complexity, the scenes encompass both indoor and outdoor environments and typically include intricate objects such as plants and flowers. In terms of challenges, many scenes were captured with loops, including large loops that end with the starting point and smaller loops within the capture process. This dataset is specifically designed for SLAM tasks and presents a high level of complexity.

## B.7 DETAILS ON OPEN-SET LOCALIZATION EXPERIMENTS

We perform 3D annotations on Replica [21], ScanNet [122], and our in-the-wild data (Fig. 11). The 3D bounding boxes for Replica and ScanNet are obtained from the official Replica dataset and EDA [141], respectively, with some objects being re-annotated by us. For our data, all 3D bounding boxes are annotated by us. Each dataset is annotated with approximately 50 objects. To ensure the thoroughness and challenge of our experiments, we design complex textual descriptions for the objects, including common names, abstract uses, colors, and other visual attributes of the objects. Examples on three datasets are presented in Fig. 11.

We employ the following method for our experiments. For a given text input, we first convert it into a feature using CLIP and then compute the similarity between this feature and the Gaussians in the scene. We select a group of Gaussians with the highest similarity. Based on their coordinates, we can derive a predicted 3D bounding box. Subsequently, we use point cloud registration method [142] to align this group of Gaussians with the 3D ground truth scene. Finally, we calculate the overlap between the predicted bounding box and the annotated object bounding box.

To evaluate the accuracy of our predictions, we use two metrics: acc@0.25 and acc@0.5. These metrics measure the percentage of predicted bounding boxes that have an IoU greater than or equal to 0.25 and 0.5, respectively, when compared to the ground truth bounding boxes.

## B.8 MORE DIAGNOSTIC EXPERIMENTS

Experiments are also on *Room 0* of Replica [21] with the baseline of SplaTAM [20], as in §4.5.

**Mapping Frame Selection.** We conduct an experiment without mapping frame selection, showing that it improves scene reconstruction quality and camera pose estimation, as well as enhancing the accuracy of the semantic field.

|  | PSNR ↑ | ATE RMSE ↓ | mIoU ↑ |
|---|---|---|---|
| *w/o* MFS | 34.52 | 0.26 | 91.57 |
| **Ours** | **34.86** | **0.24** | **96.93** |

Table 11: **Quantitative Results of Mapping Frame Selection** (MFS) (*supp.* §B.8). MFS represents mapping frame selection.

**More Details on Large-Scale Scenes Diagnostic Experiments.** We select SGS-SLAM [64] and GS³LAM [67] as baselines because they are both SplaTAM-based methods, enabling more direct comparisons. We generate labels using the methods mentioned on the Github page of GS³LAM.

**Memory Footprint of Storing the Scene.** Table 12 gives the memory requirements of *Room 0* with different $D_s$ w/ or w/o the feature pool. When assigning a 16-dimensional feature to each Gaussian, the experiment fails due to the excessive GPU memory occupancy,

|  | $D_s$ | Storage (MB) ↓ |
|---|---|---|
| SplaTAM [20] | - | 252.89 |
| *w/o $F_p$* | 3 | **317.06** |
|  | 5 | 351.14 |
|  | 16 | ~650 |
| *w/ $F_p$* | 3 | 317.79 |
|  | 5 | **318.60** |
|  | 16 | **321.95** |

Table 12: **Memory Footprint** (*supp.* §B.8) of storing *Room 0* of Replica [21].

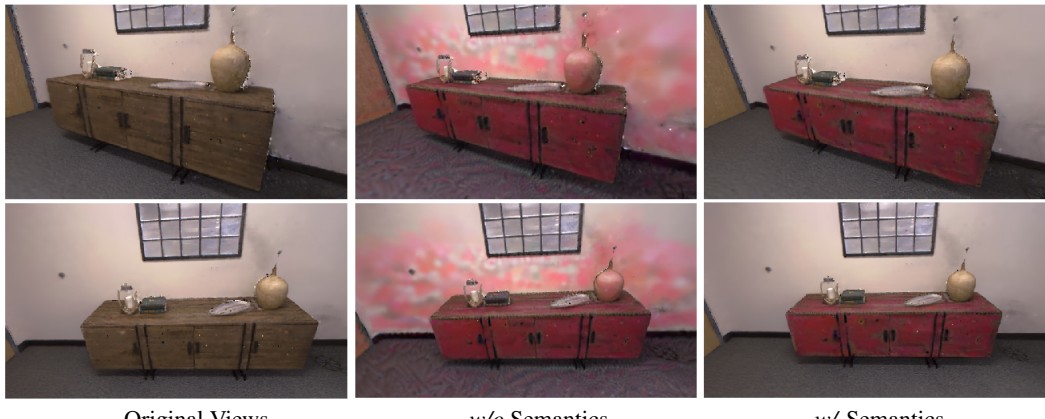

|  Original Views | *w/o* Semantics | *w/* Semantics |

Figure 12: **Diagnostic Experiment on Semantic Features for 3D Editing** (*supp.* §B.8). Prompt: *Turn the cabinet red.*

and the number is estimated. Because of the additional attribute ($k$ or the semantic feature for each Gaussian), our method requires more storage compared to SplaTAM [20]. Without $F_p$, storage demand increases greatly with higher $D_s$, that using a 5-dimensional feature requires more storage (30 MB) than a 16-dimensional $F_p$. The results illustrate that our method can substantially reduce storage requirements.

**Number of Nearest Neighbors ($m$) in Aggregation.** In Table 13, we discuss the number of nearest neighbors to select during the aggregation. When $m = 1$, the approach essentially stores semantics directly in the pool rather than aggregating them. The results show poor semantic field reconstruction, demonstrating the effectiveness of our aggregation strategy. As $m$ increases, the performance improves significantly, indicating that aggregating more features is beneficial for semantic representation.

| $m$ | mIoU ↑ |
|---|---|
| 1 | 58.12 |
| 3 | 90.66 |
| **5** | **96.93** |
| 10 | 96.84 |

Table 13: $m$ **in Aggregation** (*supp.* §B.8).

**Novel Feature Determination Threshold During Pool Expansion.** When a semantic feature is input, its similarity with features in $F_p$ is computed to determined whether it is a novel feature with a threshold (§3.2.1). In this section, we conduct an ablation study on it. As shown in Table 14, the mIoU peaks when the threshold is set to 0.8. Based on this finding, we adopt this value as the default setting for all other experiments.

| Threshold | mIoU ↑ |
|---|---|
| 0.6 | 81.67 |
| 0.7 | 90.53 |
| **0.8** | **96.93** |
| 0.9 | 95.26 |

Table 14: **Determination Threshold** (*supp.* §B.8).

**Determination of Empty Slots in $F_p$.** To determine if there are available slots in $F_p$, we calculate the similarity between each semantic in current $F_p$ and its initial state. If the similarity of all slots in $F_p$ is below a certain threshold, the current $F_p$ is considered full and will be expanded (§3.2.1). In this section, we conduct ablation experiments on this threshold. The results are shown in Table 15, including both mIoU performance and the final pool size. When the threshold is too low, some slots that may have already stored substantial semantic information are still judged as empty, causing excessive semantics to be crammed into the pool and resulting in poor performance. When it is too high, the final $F_p$ becomes excessively large, leading to a sharp increase in computational and storage

| Threshold | mIoU ↑ | $L$ |
|---|---|---|
| 0.3 | 89.64 | 50 |
| 0.4 | 92.15 | 100 |
| **0.5** | **96.93** | 200 |
| 0.6 | - | >800 |

Table 15: **Determination of Empty Slots in $F_p$** (*supp.* §B.8). When the threshold reaches 0.6, the final size exceeds 800, causing the experiment to fail.

costs. Notably, current experiments are conducted based on SplaTAM. As mentioned in *supp.* §B.4, SplaTAM's reconstructed scenes contain an excessive number of Gaussians, which consumes substantial computational resources during semantic assignment. When more compact Gaussian representations are employed, the pool can be further expanded in size, and can store more semantics in large-scale scenes.

**Expansion Proportion ($n$) of Semantic Pool.** We further explore the impact of enlarging the pool size on performance in Table 16. Results show that the greater the enlargement, the more pronounced the performance enhancement. This is attributed to the larger new space offering more flexibility for the aggregation of new semantics, thereby storing new semantics in the extended part rather than continuing to aggregate it within the original portion.

| $n$ | mIoU ↑ |
|---|---|
| 1.25 | 90.76 |
| 1.50 | 91.88 |
| 1.75 | 93.24 |
| **2.00** | **96.93** |

Table 16: $n$ **of Semantic Pool** (*supp.* §B.8).

**Semantic Features for 3D Editing.** Fig. 12 shows that, without semantics, editing operations tend to affect the entire scene, which is particularly problematic in large-scale environments. With semantics, our method enables selective refinement of targeted objects while preserving the surrounding scene. This highlights the importance of semantics in ensuring precise edits. The results further demonstrate the accuracy of the reconstructed semantic fields.

## C  MORE DISCUSSION

### C.1  DETAILED FRAMEWORK ALGORITHM

Algorithm 1 provides the pseudocode for our framework. By leveraging the expandable semantic pool, the Intra-Inter Semantic Consistency Objective and Semantic Stability Guidance, our approach integrates an expandable open-set semantic field reconstruction into SLAM, facilitating its application across various tasks.

### C.2  MORE DISCUSSIONS ON POOL EXPANSION IN LARGE SCENES

We acknowledge that for extremely large-scale scenes, the semantic pool may continue to expand and could eventually approach practical limits imposed by hardware capacity and computational resources. Neverthe-less, we believe our method remains

| Scenes | Total Frames | Expansion Frame IDs | Final Pool Size |
|---|---|---|---|
| Replica Room 0 | 2000 | 16, 22, 118 | 200 |
| TUM Long Office | 2585 | 2, 5, 180 | 200 |
| ScanNet 0000 | 5578 | 119, 347, 645, 4292 | 400 |
| In-the-wild Data | 8501 | 168, 1041, 4569 | 200 |

Table 17: **Statistics of semantic pool expansion** across relatively large-scale scenes. (*supp.* §C.2).

applicable beyond the scene scales demonstrated in this paper. As shown in Table 17, we observe a self-limiting expansion behavior of the semantic pool: while semantic diversity is continuously accumulated as the scene grows, the frequency of pool growth decreases significantly over time. This suggests that as the semantic coverage of the scene becomes increasingly comprehensive, the representation gradually stabilizes rather than expanding indefinitely, indicating strong potential for scaling to even larger environments.

## D  LEGAL AND ETHICAL CONSIDERATIONS

### D.1  ASSET LICENSE

We conduct our method on three indoor datasets (*e.g.*, Replica [21], TUM-RGBD [121] and ScanNet [122]), and two pretrained models (*e.g.*, SAM [12] and CLIP [11]), which are all available for academic access. Replica (https://github.com/facebookresearch/Replica-Dataset) is released under this License. TUM-RGBD (https://cvg.cit.tum.de/data/datasets/rgbd-dataset) is released under this License. ScanNet (http://www.scan-net.org/) is released under this License. SAM (https://segment-anything.com/) is released under this License. CLIP (https://openai.com/index/clip/) is released under this MIT License.

### D.2  POTENTIAL SOCIETAL IMPACT

Enabling everyday devices to create interactive metric-semantic 3D virtual worlds has long been a goal of AI research. We have made significant progress toward this by allowing SLAM frameworks to reconstruct open-ended semantic fields in any scene. However, this raises privacy and legal concerns. We recommend that users and content publishers restrict the use of this system to public environments to protect privacy and ensure compliance with laws, such as portrait rights and trademark protections.

### D.3  USAGE OF LLM

During paper writing, we use LLM to polish our text and check for spelling and grammatical errors.

---

**Algorithm 1** Framework Pseudocode, PyTorch-like style

---

```python
# G: Gaussians from the scene
# R: Gaussian renderer
# T, M: number of tracking and mapping iterations
# L1, SSIM: mean absolute error and structure similarity index measure

# load every timestep frame from dataset
# c, d, s, s_p: current RGB, depth, semantic frames and previous semantic frames
# K_p, F_p: key pool and feature pool
# cam: estimated cameras
# t1, t2, n: thresholds during expansion

K_p, F_p = torch.rand(L, Dk), torch.rand(L, Ds)
init_F_p = F_p.clone()

def EXPAND(s, K_p, F_p, init_F_p):
  unique_features = torch.unique(s)

  # Compute the number of novel features
  novel_feature_num = 0
  for unique_feature in unique_features:
    top_m_similarities = top_m(cosine_similarity(unique_feature, F_p))
    if all(top_m_similarities > t1):
      novel_feature_num += 1

  # Check if there is enough slots to store novel features
  empty_slots = torch.sum(cosine_similarity(F_p, init_F_p) < t2)

  if novel_feature_num > empty_slots:
      # Expand K_p and F_p with randomly initialized parts
      expand_K_p, expand_F_p = torch.rand((n - 1)L, Dk), torch.rand((n - 1)L, Ds)
      K_p, F_p = torch.cat(K_p, expand_K_p), torch.cat(F_p, expand_F_p)
      init_F_p = torch.cat(init_F_p, expand_F_p)

  return K_p, F_p, init_F_p

for c, d, s in data_loader:
  # Tracking
  for _ in range(T):
    # Assign semantic features to Gaussians (Eq.5)
    G.params.semantic = FEATURE_POOL(G.params.key, K_p, F_p)

    # Render frames (Eqs.3,4,6)
    c_t, d_t, s_t = R(G.params, cam.params)

    # Compute tracking loss (Eqs.7-8)
    loss_t_c = L1(c, c_t)
    loss_t_d = L1(d, d_t)
    loss_t_s = L1(s, s_t)
    loss_t = loss_t_c + loss_t_d + loss_t_s

    # Update the cameras parameters
    loss_t.backward()
    update(cam.params)

  # Mapping
  for _ in range(M):
    K_p, F_p, init_F_p = EXPAND(s, K_p, F_p, init_F_p)

    G.params.semantic = FEATURE_POOL(G.params.key, K_p, F_p)

    c_m, d_m, s_m = R(G.params, cam.params)

    # Compute RGB and depth loss (Eqs.11-12)
    loss_m_c = L1(c, c_m) + SSIM(c, c_m)
    loss_m_d = L1(d, d_m)

    # Intra-Inter Semantic Consistency Objective & Semantic Stability Guidance
    loss_m_co = CONSISTENCY_OBJECTIVE(s_m, s_p)
    m_sg = STABILITY_GUIDANCE(s_m, s_p)

    # Semantic loss (Eq.13)
    loss_m_s = m_sg * L1(s, s_m) + loss_m_co

    # Overall mapping loss (Eq.14)
    loss_m = loss_m_c + loss_m_d + loss_m_s

    # Update the Gaussian parameters
    loss_m.backward()
    update(G.params)
```

---

| Methods | TUM-RGBD [121] | | | | | | ScanNet [122] | | | | | | |
|---|---|---|---|---|---|---|---|---|---|---|---|---|---|
| | Avg. | fr1/ desk | fr1/ desk2 | fr1/ room | fr2/ xyz | fr3/ off. | Avg. | 0000 | 0059 | 0106 | 0169 | 0181 | 0207 |
| Vox-Fusion [43] | 11.31 | 3.52 | 6.00 | 19.53 | 1.49 | 26.01 | 26.90 | 68.84 | 24.18 | 8.41 | 27.28 | 23.30 | 9.41 |
| NICE-SLAM [41] | 15.87 | 4.26 | 4.99 | 34.49 | 31.73 | 3.87 | 10.70 | 12.00 | 14.00 | 7.90 | 10.90 | 13.40 | **6.20** |
| Point-SLAM [18] | 8.92 | 4.34 | 4.54 | 30.92 | 1.31 | 3.48 | 12.19 | 10.24 | 7.81 | 8.65 | 22.16 | 14.77 | 9.54 |
| GS³LAM [67] | - | - | - | - | - | - | 11.96 | 15.23 | 9.87 | 15.56 | 12.98 | 10.54 | 7.56 |
| SplaTAM [20] | 5.48 | 3.35 | 6.54 | 11.13 | **1.24** | 5.16 | 11.88 | 12.83 | 10.10 | 17.72 | 12.08 | 11.10 | 7.46 |
| **SplaTAM + Ours** | 5.20$_{\downarrow 0.28}$ | 3.18 | 6.48 | 10.95 | _1.29_ | 4.12 | 10.27$_{\downarrow 1.61}$ | 14.56 | 9.41 | 7.82 | 11.59 | 11.11 | 7.12 |
| LoopSplat [19] | _3.33_ | _2.08_ | _3.54_ | _6.24_ | 1.58 | _3.22_ | _7.73_ | 6.20 | _7.10_ | **7.40** | _10.60_ | _8.50_ | 6.60 |
| **LoopSplat + Ours** | **3.13**$_{\downarrow 0.20}$ | **1.94** | **3.32** | **6.05** | 1.37 | **2.99** | **7.54**$_{\downarrow 0.19}$ | 6.12 | 7.02 | _7.41_ | 9.89 | 8.31 | _6.48_ |

Table 18: **Quantitative Results on Camera Pose Estimation** (*supp.* §B.2.2) on TUM-RGBD [121] and ScanNet [122] (ATE RMSE↓ [cm]).

| Methods | Metrics | TUM-RGBD [121] | | | | | | ScanNet [122] | | | | | | |
|---|---|---|---|---|---|---|---|---|---|---|---|---|---|---|
| | | Avg. | fr1/ desk | fr1/ desk2 | fr1/ room | fr2/ xyz | fr3/ off. | Avg. | 0000 | 0059 | 0106 | 0169 | 0181 | 0207 |
| Vox-Fusion [43] [ISMAR22] | PSNR↑ | 15.54 | 15.79 | 14.12 | 14.20 | 16.32 | 17.27 | 18.17 | 19.06 | 16.38 | _18.46_ | 18.69 | 16.75 | 19.66 |
| | SSIM↑ | 0.632 | 0.647 | 0.568 | 0.566 | 0.706 | 0.677 | 0.673 | 0.662 | 0.615 | _0.753_ | 0.650 | 0.666 | 0.696 |
| | LPIPS↓ | 0.502 | 0.523 | 0.541 | 0.559 | 0.433 | 0.456 | 0.504 | 0.515 | 0.528 | 0.439 | 0.513 | 0.532 | 0.500 |
| NICE-SLAM [41] [CVPR22] | PSNR↑ | 13.59 | 13.83 | 12.00 | 11.39 | 17.87 | 12.89 | 17.54 | 18.71 | 16.55 | 17.29 | 18.75 | 15.56 | 18.38 |
| | SSIM↑ | 0.545 | 0.569 | 0.514 | 0.373 | 0.718 | 0.554 | 0.621 | 0.641 | 0.605 | 0.646 | 0.629 | 0.562 | 0.646 |
| | LPIPS↓ | 0.494 | 0.482 | 0.520 | 0.629 | 0.344 | 0.498 | 0.548 | 0.561 | 0.534 | 0.510 | 0.534 | 0.602 | 0.552 |
| ESLAM [47] [CVPR23] | PSNR↑ | 13.42 | 11.29 | 12.30 | 9.06 | 17.46 | 17.02 | 15.29 | 15.70 | 14.48 | 15.44 | 14.56 | 14.22 | 17.32 |
| | SSIM↑ | 0.599 | 0.666 | 0.634 | **0.929** | 0.310 | 0.457 | 0.658 | 0.687 | 0.632 | 0.628 | 0.656 | 0.696 | 0.653 |
| | LPIPS↓ | 0.464 | 0.358 | 0.421 | **0.192** | 0.698 | 0.652 | 0.488 | 0.449 | 0.450 | 0.529 | 0.486 | 0.482 | 0.534 |
| Point-SLAM [18] [ICCV23] | PSNR↑ | 15.63 | 13.87 | 14.12 | 14.16 | 17.56 | 18.43 | 19.82 | 21.30 | 19.48 | 16.80 | 18.53 | 22.27 | 20.56 |
| | SSIM↑ | 0.665 | 0.627 | 0.592 | 0.645 | 0.708 | 0.754 | 0.751 | 0.806 | 0.765 | 0.676 | 0.686 | 0.823 | 0.750 |
| | LPIPS↓ | 0.538 | 0.544 | 0.568 | 0.546 | 0.585 | 0.448 | 0.514 | 0.485 | 0.499 | 0.544 | 0.542 | 0.471 | 0.544 |
| GS³LAM [67] [ACMMM24] | PSNR↑ | - | - | - | - | - | - | 22.86 | 23.02 | 20.96 | 22.37 | 25.85 | 20.58 | 24.39 |
| | SSIM↑ | - | - | - | - | - | - | _0.868_ | _0.852_ | _0.858_ | _0.872_ | _0.890_ | _0.855_ | _0.878_ |
| | LPIPS↓ | - | - | - | - | - | - | _0.222_ | 0.277 | _0.213_ | _0.205_ | 0.189 | _0.252_ | 0.195 |
| SplaTAM [20] [CVPR24] | PSNR↑ | 21.67 | 22.00 | 20.35 | 19.62 | _24.50_ | 21.90 | 19.14 | 19.33 | 19.27 | 17.73 | 21.97 | 16.76 | 19.80 |
| | SSIM↑ | 0.856 | 0.857 | 0.788 | 0.812 | _0.947_ | 0.876 | 0.716 | 0.660 | 0.792 | 0.690 | 0.776 | 0.683 | 0.696 |
| | LPIPS↓ | 0.208 | 0.232 | 0.270 | 0.235 | _0.100_ | 0.202 | 0.358 | 0.438 | 0.289 | 0.376 | 0.281 | 0.420 | 0.341 |
| **SplaTAM + Ours** | PSNR↑ | **22.63**$_{\uparrow 0.96}$ | **23.68** | **21.55** | **20.89** | **24.77** | **22.28** | 22.10$_{\uparrow 2.96}$ | 21.15 | 20.43 | 21.90 | 24.89 | 19.99 | 24.26 |
| | SSIM↑ | **0.895**$_{\uparrow 0.039}$ | **0.920** | **0.873** | 0.840 | **0.950** | _0.893_ | 0.836$_{\uparrow 0.120}$ | 0.764 | 0.840 | 0.854 | 0.863 | 0.819 | 0.874 |
| | LPIPS↓ | **0.165**$_{\downarrow 0.042}$ | **0.138** | **0.185** | _0.226_ | **0.096** | **0.182** | _0.249_$_{\downarrow 0.109}$ | 0.314 | 0.231 | 0.327 | 0.222 | 0.297 | 0.204 |
| LoopSplat [19] [3DV25] | PSNR↑ | 21.30 | 22.03 | 19.60 | 18.71 | 22.68 | _23.47_ | _24.92_ | _24.99_ | _23.23_ | _23.35_ | _26.80_ | _24.82_ | _26.33_ |
| | SSIM↑ | 0.837 | 0.849 | 0.773 | 0.791 | 0.892 | 0.879 | 0.845 | 0.840 | 0.831 | 0.846 | 0.877 | 0.824 | 0.854 |
| | LPIPS↓ | 0.297 | 0.307 | 0.386 | 0.324 | 0.217 | 0.253 | 0.425 | 0.450 | 0.400 | 0.409 | 0.346 | 0.514 | 0.430 |
| **LoopSplat + Ours** | PSNR↑ | _22.25_$_{\uparrow 0.95}$ | _22.99_ | _20.37_ | _20.15_ | 23.61 | **24.13** | **25.53**$_{\uparrow 0.61}$ | **25.30** | **24.11** | **23.98** | **27.41** | **25.34** | **27.06** |
| | SSIM↑ | _0.893_$_{\uparrow 0.056}$ | _0.887_ | _0.864_ | _0.881_ | 0.928 | **0.905** | **0.896**$_{\uparrow 0.060}$ | **0.884** | **0.876** | **0.901** | **0.918** | **0.885** | **0.913** |
| | LPIPS↓ | _0.220_$_{\downarrow 0.077}$ | _0.191_ | 0.284 | 0.253 | 0.140 | 0.231 | 0.305$_{\downarrow 0.120}$ | 0.325 | 0.301 | 0.316 | 0.250 | 0.349 | 0.288 |

Table 19: **Quantitative Comparison on Rendering Performance** (*supp.* §B.2.2) with baselines on TUM-RGBD [121] and ScanNet [122].

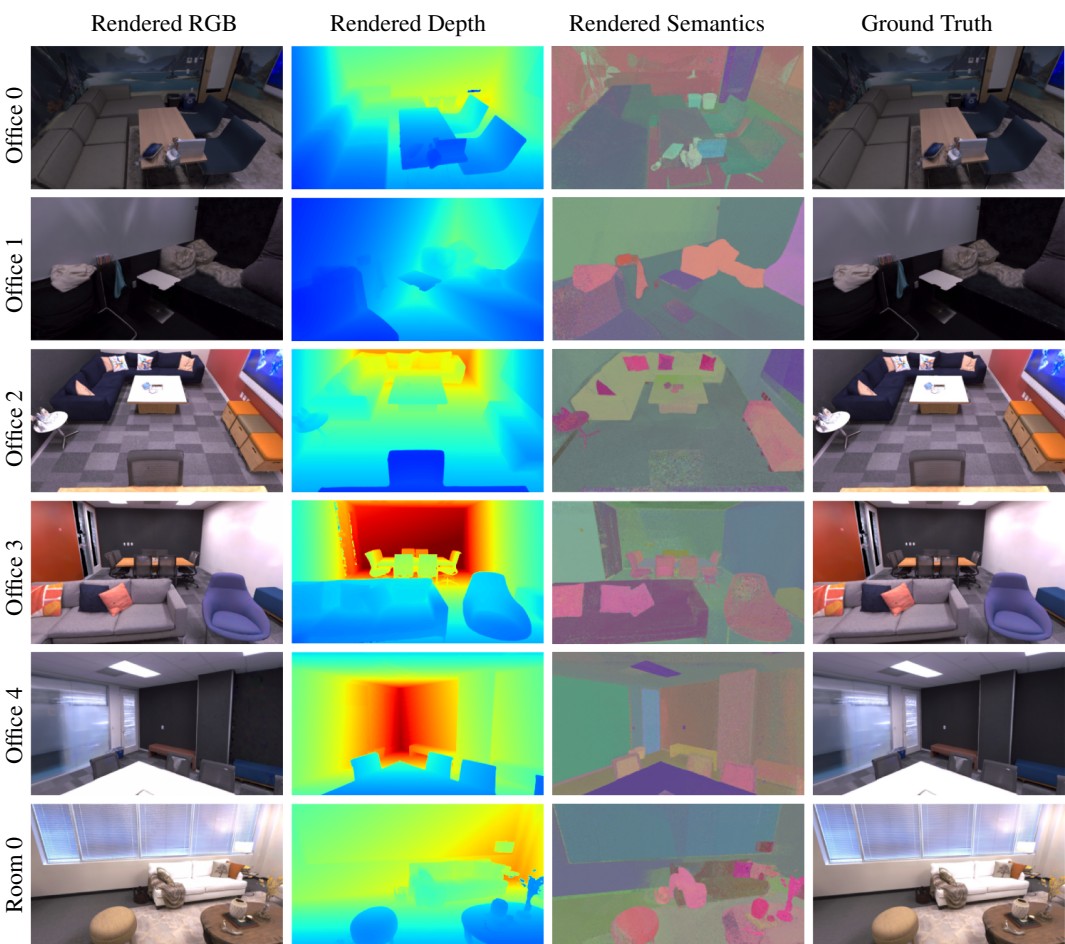

Figure 13: **More Qualitative Results** (*supp.* §B.2.1) of RGB, depth and open-set semantics on Replica [21].

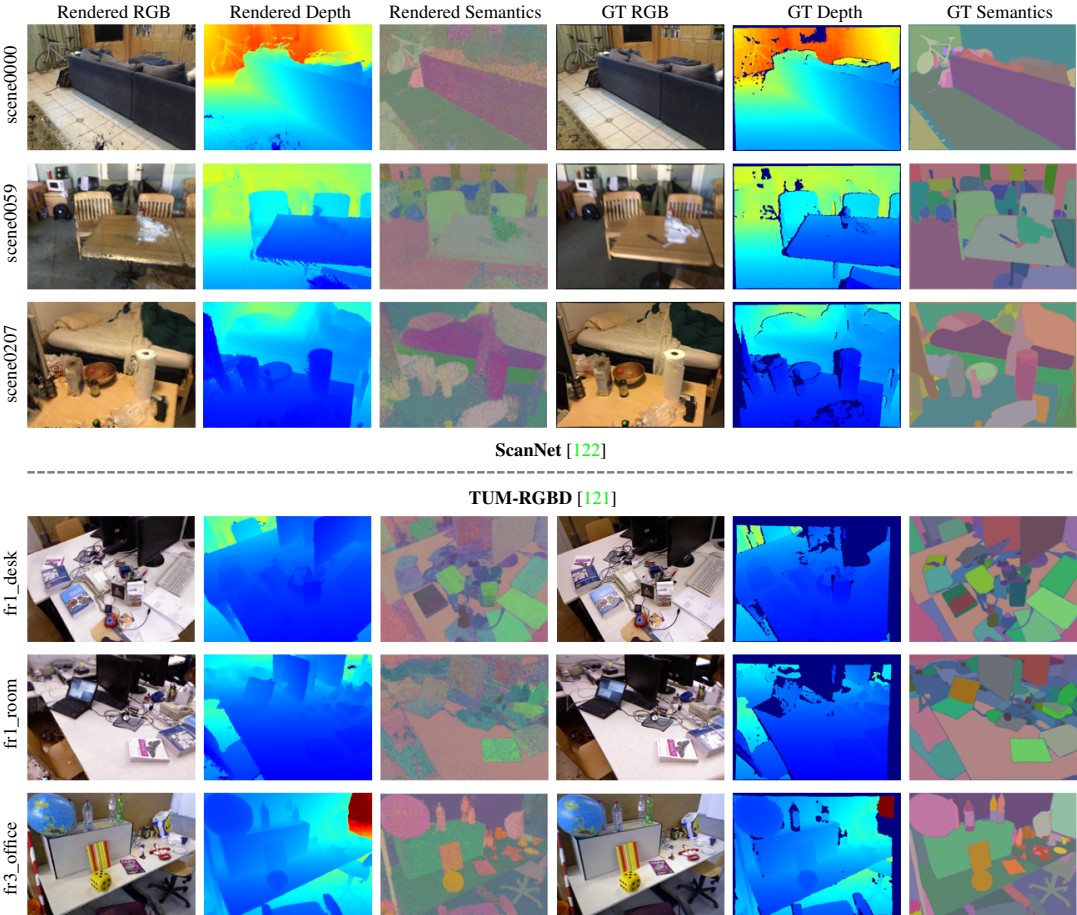

Figure 14: **Qualitative Results** (*supp.* §B.2.2) of RGB, depth and open-set semantics on ScanNet [122] and TUM-RGBD [121].

