# OpenReview forum: "Open-Set Semantic Gaussian Splatting SLAM with Expandable Representation"
_ICLR.cc/2026/Conference — ICLR 2026 Poster_

### Official Review · Reviewer_fyiJ · 2025-10-26

**Soundness:** 4
**Presentation:** 4
**Contribution:** 4
**Rating:** 8
**Confidence:** 5

**Summary:**

The paper, "Open-Set Semantic Gaussian Splatting SLAM with Expandable Representation“, introduces a system designed to enable everyday devices like smartphones to capture and reconstruct 3D scenes with rich, open-set semantics. The core idea is to integrate an expandable semantic feature pool with a Gaussian Splatting SLAM (GS-SLAM) framework. This feature pool decouples scene-level semantics from individual 3D Gaussians, using lightweight indexing vectors to reduce memory overhead and support dynamic updates of semantics. The system also incorporates a consistency-aware optimization strategy and a Semantic Stability Guidance mechanism to improve cross-view semantic consistency.

**Strengths:**

The paper presents a compelling and timely contribution at the intersection of dense SLAM, 3D Gaussian splatting, and open-vocabulary semantic understanding.

The work demonstrates high originality through several key innovations:

1. Expandable Semantic Feature Pool: Rather than embedding high-dimensional semantic features directly into each Gaussian (which is memory-prohibitive), the authors propose a shared, dynamic, and expandable semantic feature pool. Each Gaussian references semantics via a lightweight indexing vector. This design is both novel and pragmatic—it decouples scene-level semantics from per-point storage, enabling scalability and dynamic updates.

2. Open-Set Semantic Integration in SLAM: While prior SLAM systems typically handle closed-set semantics (e.g., fixed object categories), this work enables open-set semantic understanding—supporting queries like “fruit in plastic bag” or “xbox”—by leveraging foundation models (e.g., CLIP) within a real-time SLAM pipeline. This bridges a critical gap between foundation model capabilities and real-world 3D reconstruction.

3. Consistency-Aware Optimization: The paper introduces a dual mechanism—an Intra-Inter Semantic Consistency Objective (via contrastive learning) and Semantic Stability Guidance (via cosine-similarity-based reweighting)—to enforce cross-view and temporal semantic coherence. This addresses a fundamental limitation of naively lifting 2D foundation model outputs into 3D without considering geometric consistency.

**Weaknesses:**

1. While the paper claims to reduce memory overhead "by orders of magnitude" (line 037) through the expandable semantic feature pool, the memory footprint analysis in Table 5 and Table 10 still shows significant memory usage, particularly as the semantic feature dimension ($D_s$) or pool size ($L$) increases. For real-time applications on "everyday devices (e.g., smartphones)" (line 025), the current memory requirements might still be prohibitive.

2. The runtime analysis in Table 8 shows that the proposed method incurs an "approximately 8% – 21% additional time overhead" compared to SplaTAM and LoopSplat. While this might be acceptable for some applications, achieving truly real-time performance on mobile devices for complex scenes with continuous semantic updates remains a challenge. The paper mentions that the "additional computational overhead stems from semantic assignment from Fp" and the consistency objectives.

3. The paper states that the system "demonstrates limited robustness in highly dynamic or large-scale (e.g., city-level) environments" (lines 1150-1151). This is a significant limitation for a SLAM system designed for "in-the-wild 3D scenes" (line 026). While the Semantic Stability Guidance mechanism is introduced to mitigate semantic ambiguity, a more thorough analysis of its effectiveness in highly ambiguous or dynamic scenarios would be beneficial.

4. The system relies on pre-trained 2D foundation models like CLIP [11] or SAM [12] for semantic distillation. While this is a common approach, the performance and generalizability of the system will be inherently tied to the capabilities and biases of these underlying models. The paper mentions that the framework is not dependent on "any specific semantic extraction method" (line 826), but the choice of these models can still impact the quality of the semantic representations.

**Questions:**

See the weaknesses.

---

> ### Author Response · Authors · 2025-11-22
> **Point-to-Point Response to Reviewer fyiJ**
>
> We thank reviewer fyiJ for the valuable time and constructive feedback. We provide point-to-point response below.
>
> **Q1: Discussion on reduced memory overhead.**
>
> **A1:** Keen observation! However, to our best knowledge, real-time dense semantic SLAM on everyday devices remains extremely challenging. Our contribution focuses on reducing the semantic memory bottleneck, which is the primary obstacle preventing prior works from even approaching such scenarios.
>
> + As shown in Table 5, without a pool, the system fails even at 6-dim features due to memory blow-up, whereas with a pool we can run **64**-dim semantics stably, with a memory increase of only a few GB on desktop GPUs.
> + Table 10 shows that with a feature pool, memory footprint increases memory by **1.31%** when increasing semantic dimensions from 3-dim to 16-dim. Whereas without the pool, this increase is ~105%.
>
> These results collectively validate our method brings the semantic component down to a footprint compatible with future optimization of the rest of the SLAM pipeline.
>
> **Q2: More discussion on runtime.**
>
> **A2:** Thank you for the thoughtful feedback.
> We agree that achieving real-time performance remains challenging for dense visual SLAM systems, and our current implementation is not yet fully real-time. In this context, our goal is to make semantic processing no longer the limiting factor, and the current results show that the additional computational overhead is closely related to the number of Gaussians involved in semantic optimization, thus reducing this number yields a clear decrease in runtime.
>
> To improve efficiency, we plan to adopt two complementary strategies. **First**, a hierarchical Gaussian representation that enables coarse-to-fine semantic assignment, ensuring that only the most relevant Gaussians at appropriate levels participate in semantic optimization. **Second**, semantic-guided pruning and fusion, which leverages semantic consistency to remove redundant Gaussians and merge homogeneous regions, thereby reducing both the total number of Gaussians and the per-iteration computational cost.
>
> **Q3: More discussion on robustness of Semantic Stability Guidance mechanism in highly dynamic or large-scale environments.**
>
> **A3:** Good suggestion! In our paper, "highly dynamic scenes" refer to settings with strong, frequent object motion, and "large-scale scenes" refer to community- or city-level scenarios spanning long trajectories and wide areas. Our statement that the system "demonstrates limited robustness in highly dynamic or large-scale environments" is intended as a clarification of the current scope of our work rather than a fundamental limitation of the Semantic Stability Guidance mechanism itself.
>
> However, we believe Semantic Stability Guidance is compatible with and potentially beneficial for these more challenging settings:
>
> + For dynamic scenes, recent research [ref1-4] has begun exploring the modeling of dynamic foreground objects. The cross-frame correspondence capability holds the potential to offer new insights for the field of dynamic SLAM, though direct application still presents challenges.
> + For large-scale / city-level scenes, the Semantic Stability Guidance is not negatively impacted by increasing scene scale; on the contrary, as the number of reference frames grows, its capability to determine correct semantics becomes stronger.
>
> [ref1] DG-SLAM: Robust Dynamic Gaussian Splatting SLAM with Hybrid Pose Optimization. NeurIPS 2024.
>
> [ref2] WildGS-SLAM: Monocular Gaussian Splatting SLAM in Dynamic Environments. CVPR 2025.
>
> [ref3] DyGS-SLAM: Real-Time Accurate Localization and Gaussian Reconstruction for Dynamic Scenes. ICCV 2025.
>
> [ref4] 4D Gaussian Splatting SLAM. ICCV 2025.
>
> **Q4: "The choice of fundamental models can still impact the quality of the semantic representations."**
>
> **A4:** Thanks for the insightful comment. Our statement that the framework is "not dependent on any specific semantic extraction method" means that the 3D reasoning and reconstruction components are modular and can be paired with any semantic extractor, rather than being tailored to CLIP or SAM in particular.
>
> In our experiments, we adopt SAM+CLIP to ensure a fair and direct comparison with prior works such as LERF [27] and GOI [17]. Importantly, this design choice makes our method future-proof: as stronger or less biased foundation models become available, they can be plugged into our pipeline without modifying the core method, thereby improving the semantic representation quality. Therefore, we argue that this flexibility constitutes an advantage rather than a weakness.

---

> > ### Comment · Reviewer_fyiJ · 2025-11-25
> >
> > Thanks for the update. No further questions from me.

---

### Official Review · Reviewer_r2wC · 2025-10-31

**Soundness:** 2
**Presentation:** 2
**Contribution:** 2
**Rating:** 4
**Confidence:** 4

**Summary:**

The paper presents a Gaussian-based SLAM framework that integrates open-vocabulary semantic features derived from pretrained visual-language models. It introduces an updating mechanism for semantic features across frames, aiming to improve temporal consistency and robustness in semantic mapping. The method is evaluated on multiple RGB-D datasets, showing improvements in both geometric accuracy and semantic segmentation performance.

**Strengths:**

- Leveraging pretrained models for enhancing scene-level semantic awareness is meaningful and practically relevant for scalable 3D mapping.

- The integration of semantic information into a Gaussian-based SLAM framework is valuable and timely, addressing the growing need for open-vocabulary 3D understanding on real-time systems.

- The experimental section covers multiple datasets and tasks， typically three in-the-wild scenes captured by everyday devices are interesting.

- The visualization in the paper is clear and informative, effectively illustrating the semantic reconstruction results and helping understand the pipeline of the proposed method.

**Weaknesses:**

- The quantitative comparison in Table 1 lacks several recent SOTA RGB-D SLAM baselines such as MonoGS, Gaussian-SLAM, RTG-SLAM, and SplatSLAM. It is highly recommended to include these baselines to ensure a fair, comprehensive, and up-to-date evaluation of the proposed module's performance.

- The organization of the paper needs substantial improvement. Essential information about how semantic features are extracted, processed, and integrated into the pipeline is relegated to Appendix A.1, while the main body spends excessive space on general preliminaries and related work (Eq. 1–5, 10–11). As a result, readers struggle to understand the core contribution without repeatedly consulting the supplementary material. The authors should move the key descriptions of the semantic feature pipeline, backbone choice, 2D-to-3D mapping, update rules, and training losses into the main Method section, and condense redundant preliminary equations.

- The use of pretrained semantic features from CLIP and DINOv2 is no longer novel, as several recent works, such as SemGauss-SLAM and OVO-SLAM, have already demonstrated open-vocabulary mapping with similar embeddings. The only distinctive component here is the recurrent update mechanism applied to explicit per-map semantic features. However, the motivation for updating explicit feature representations across frames is not clearly justified. Updating/Memorization is intuitive for latent representations that capture temporal dependencies, but less so for fixed explicit map representations, as per pixel level, their cross-frame correspondence is not recurrently updated but shifted from adjacent pixels in a continuous trajectory.

**Questions:**

Please refer to the weaknesses.

---

> ### Author Response · Authors · 2025-11-22
> **Point-to-Point Response to Reviewer r2wC (1/2)**
>
> We thank reviewer r2wC for the valuable time and constructive feedback. We provide point-to-point response below.
>
> **Q1: Extra comparisons with more baselines.**
>
> **A1:** Thanks for the suggestion. Regarding the mentioned works: Gaussian-SLAM [45] and Splat-SLAM [50] are not yet accepted to any conference or journal; RTG-SLAM [54], while promising, has been shown to perform inferior to LoopSplat [19], which is a more recent and well-established method, and is already included as a baseline in our main paper.
>
> To further address your concern, we additionally include comparisons against MonoGS [21], S3PO-GS [ref1], and SEGS-SLAM [ref2]. MonoGS [21] is a representative early 3DGS-based SLAM system; S3PO-GS [ref1] and SEGS-SLAM [ref2] (both ICCV 2025) represent the most recent generation of 3DGS-based SLAM frameworks.
>
> The results below are evaluated on Replica and show that our semantic module consistently improves the underlying SLAM performance.
>
> | Methods | ATE RMSE $\downarrow$ | PSNR $\uparrow$ | SSIM $\uparrow$ | LPIPS $\downarrow$ |
> |:---------:|:---:|:---:|:---:|:---:|
> | MonoGS [21] [CVPR24] | 0.79 | 37.50 | 0.960 | 0.070 |
> | **MonoGS + Ours** | **0.55** | **38.39** | **0.976** | **0.049** |
> | LoopSplat^ [19] [3DV25] | 0.26 | 36.63 | 0.985 | 0.112 |
> | **LoopSplat + Ours** | **0.23** | **38.05** | **0.989** | **0.051** |
> | S3PO-GS^^ [ref1] [ICCV25] | 2.26 | 32.71 | 0.943 | 0.138 |
> | **S3PO-GS + Ours** | **0.97** | **35.23** | **0.961** | **0.070** |
> | SEGS-SLAM [ref2] [ICCV25] | 0.43 | 39.42 | 0.975 | 0.021 |
> | **SEGS-SLAM + Ours** | **0.38** | **39.88** | **0.983** | **0.020** |
>
> ^ LoopSplat (3DV25) is already included in our main text as a strong recent baseline.
>
> ^^ S3PO-GS targets open-world outdoor settings; thus the Replica results shown here are from our reproduction.
>
> [ref1] Outdoor Monocular SLAM with Global Scale-Consistent 3D Gaussian Pointmaps. ICCV 2025.
>
> [ref2] SEGS-SLAM: Structure-enhanced 3D Gaussian Splatting SLAM with Appearance Embedding. ICCV 2025.
>
> **Q2: Organization of the paper.**
>
> **A2:** Our intention is to keep the main text focused on our core contributions (the expandable semantic representation and the consistency-aware optimization strategy) while moving auxiliary components to the appendix to avoid diluting the central ideas.
>
> + **Preliminaries (§3.1).** Preliminaries are necessary because our method is built directly on top of 3DGS-based SLAM. Eqs. 1–5, 10–11 define the rendering formulations that our semantic mechanisms operate on; without these, the expandable representation and optimization strategy cannot be precisely described.
> + **Main Contributions (§3.2–§3.3).** All key components that constitute our actual method (including initialization, aggregation, and expansion of the semantic pool (§3.2) and the full system optimization (§3.3)) are presented in the main text.
> + **Auxiliary Components.** The parts moved to the supplement (e.g., semantic distillation, and the closed-set classification head in §4.3) are not core contributions, but rather implementation decisions specific to certain experiments. Including these in the main body would increase complexity without improving conceptual clarity.
> + **Other Key Descriptions.** 2D-to-3D mapping is standard in 3DGS-based SLAM and already appears in the main text (§3.1.1, §3.3). And training losses are presented in §3.3. They are summarized concisely to avoid redundancy.
>
> We agree that improving the flow between the semantic feature pipeline and the main method description will benefit readability. However, we note that other reviewers (*e.g.*, Reviewer 21bL and Reviewer zwtS) found the structure clear and logical.

---

> ### Author Response · Authors · 2025-11-22
> **Point-to-Point Response to Reviewer r2wC (2/2)**
>
> **Q3: Clarification on our novelty and motivation.**
>
> **A3:** Sorry for the confusion. We make detailed clarification below.
>
> > **Q3.1: "The use of pretrained semantic features ... is no longer novel."**
>
> **A3.1:** Our contribution is not in adopting these features, but in solving the system-level challenges that arise when integrating them into a sequential SLAM pipeline (Lines 059-084).
>
> > **Q3.2: Differences with previous works.**
>
> **A3.2:** SemGauss-SLAM is _closed-set_. Its semantic features are supervised using annotated label data, and it does not maintain open-vocabulary feature fields. OVO-SLAM still relies on label-level predictions, not explicit feature aggregation. It also lacks dynamic adaptation to new concepts, long-term cross-frame consistency modeling, and a compression mechanism for scalable semantics. Thus, our focus differs substantially from both.
>
> > **Q3.3: "The motivation for updating explicit feature representations across frames is not clearly justified."**
>
> **A3.3:** Explicit 3D semantic features still require recurrent updates due to a fundamental issue: *2D open-vocabulary models produce inherently inconsistent predictions across viewpoints*, even for the same object (Lines 071–072, 264–268). If we freeze each Gaussian’s semantic feature after a single observation, the map will accumulate contradictions and simply reflect the noise and viewpoint-dependent biases of 2D models. Our recurrent update mechanism is a necessary integration process that accumulates semantic evidence from multiple viewpoints, suppresses inconsistent predictions, and refines each Gaussian’s semantic identity toward a stable 3D representation.

---

> > ### Comment · Reviewer_r2wC · 2025-11-27
> >
> > I appreciate the detailed explanation and additional experiments provided in the authors’ response. Most of my concerns have been sufficiently addressed, with the exception of the paper’s structure. However, I acknowledge that different readers may have different perspectives, and the other reviewers find the current structure and figures satisfactory. I will raise my score accordingly if the above discussion of related works can be incorporated into the revised paper.

---

### Official Review · Reviewer_zwtS · 2025-10-31

**Soundness:** 3
**Presentation:** 4
**Contribution:** 3
**Rating:** 6
**Confidence:** 4

**Summary:**

This paper proposes an open-set semantic SLAM system based on 3D Gaussian Splatting. The key innovation is the introduction of an expandable semantic feature pool, which decouples high-level semantics from individual Gaussians. This design significantly reduces memory usage and allows for dynamic integration of new semantic concepts during SLAM operation. To address the inconsistency issue from 2D foundation models, the authors also propose a consistency-aware optimization strategy and a semantic stability guidance mechanism. Extensive experiments on both synthetic and real-world datasets demonstrate the system's advantages in rendering quality, tracking accuracy, and open-set semantic understanding tasks, such as 3D localization and scene editing.

**Strengths:**

The core idea of a learnable and expandable semantic feature pool is novel and clever. It effectively separates the scene-level semantics from the geometric representation, enabling efficient and scalable open-set learning.

The paper is structured logically, and the framework is explained step by step, making it understandable.

This work pushes the boundary of semantic SLAM from closed-set to open-set, which is a crucial step towards general-purpose 3D scene understanding. The ability to run on commodity hardware and support applications like 3D editing greatly enhances its practical value.

**Weaknesses:**

While the comparison with existing SLAM methods is comprehensive, I feel the comparison with state-of-the-art open-set 3D understanding methods (e.g., OpenMask3D, OpenScene) in terms of pure semantic segmentation accuracy is somewhat lacking.

The runtime analysis in the supplement shows a non-negligible overhead. Although the authors attribute this to the baseline, a more in-depth discussion on how to optimize the efficiency of the semantic module itself would be helpful for practical deployment.

The threshold parameters for the pool expansion (e.g., the similarity thresholds in Algorithm 1) seem crucial but their selection process is not deeply analyzed.

**Questions:**

1. The expansion factor n and the threshold for determining "empty slots" seem to be empirical. Could you discuss the sensitivity of the model's performance to these hyperparameters? Is there a risk of the pool expanding too aggressively in very large-scale scenes?
﻿
2. In the semantic stability guidance, how is the "object that appears for the first time" specifically identified? Is this based purely on the lack of correspondence in previous frames?
﻿
3. The paper demonstrates excellent results on in-the-wild data. Could you comment on the system's performance in highly dynamic environments where objects move frequently? Does the semantic pool and consistency mechanism handle such cases robustly?

---

> ### Author Response · Authors · 2025-11-22
> **Point-to-Point Response to Reviewer zwtS (1/2)**
>
> We thank reviewer zwtS for the valuable time and constructive feedback. We provide point-to-point response below.
>
> **Q1: Existing Comparisons with open-set 3D understanding methods.**
>
> **A1:** Thanks for the reminder. However, we would like to clarify that we have already included comparisons with prior 3D open-set semantic approaches (*e.g.*, LERF [27], LEGaussians [16], GOI [17]) in terms of both 2D segmentation and 3D localization (§4.4). These experiments demonstrate that our method reconstructs a significantly more accurate semantic field and enables direct interaction with the 3D Gaussian representation.
>
> Regarding the mentioned methods, they operate on existing 3D point clouds by performing semantic segmentation after the scene representation has already been reconstructed. In contrast, our method, similar to LEGaussians, focuses on jointly optimizing both geometry and semantics from multi-view images. This is a fundamentally different problem setting, and therefore direct comparison of pure segmentation accuracy is not meaningful, as also discussed in LEGaussians [16]: "However, these methods mainly address the comprehension and analysis of existing scene representations, like point clouds, rather than optimizing scene representation from multi-view images, which is the focus of our work."
>
> [ref1] OpenMask3D: Open-Vocabulary 3D Instance Segmentation. NeurIPS 2023.
>
> [ref2] OpenScene: 3D Scene Understanding with Open Vocabularies. CVPR 2023.
>
> **Q2: More discussion on runtime.**
>
> **A2:** Thanks for the suggestion. We agree that runtime is an important factor for practical deployment. As shown in Table 8, the additional computational overhead is closely related to the number of Gaussians involved in semantic optimization, thus reducing this number yields a clear decrease in runtime. To improve efficiency, we plan to incorporate two complementary strategies. First, Hierarchical Gaussian Representation: introduce a coarse-to-fine semantic assignment mechanism so that only a subset of Gaussians at appropriate levels participate in semantic optimization. Second, Semantic-Guided Pruning and Fusion: Utilize semantic consistency to prune redundant Gaussians and fuse semantically homogeneous regions, thereby reducing both the number of Gaussians and the per-iteration computational cost.
>
> **Q3: Selection process of threshold parameters.**
>
> **A3:** A detailed explanation of how these hyperparameters (*e.g.*, $t_1$​, $t_2$​, and $n$ in Algorithm 1) are selected is provided in _supp._ §B.7. Specifically, _supp._ §B.7 outlines:
>
>   + how we choose $t_1$​ based on the distribution of similarities to existing pool entries (see "Novel Feature Determination Threshold During Pool Expansion" paragraph),
>   + how we determine $t_2$ by monitoring final representation accuracy (see "Determination of Empty Slots in $F_p$" paragrap), and
>   + how we set $n$ using the expansion behavior observed in long sequences (see "Expansion Proportion of Semantic Pool" paragraph).

---

> ### Author Response · Authors · 2025-11-22
> **Point-to-Point Response to Reviewer zwtS (2/2)**
>
> **Q4: Sensitivity of the model's performance to hyperparameters & discussion on pool expanding in very large-scale scenes.**
>
> **A4:** Good questions! We address your concerns respectively below.
>
> > **Q4.1: Discussions on the sensitivity of the model's performance to hyperparameters.**
>
> **A4.1:** As mentioned in **A3**, we provide an extensive analysis of the hyperparameters involved in semantic pool expansion in _supp._ §B.7.
>
> > **Q4.2: "Is there a risk of the pool expanding too aggressively in very large-scale scenes?"**
>
> **A4.2:** We agree that uncontrolled expansion would be undesirable. Empirically, however, the pool exhibits self-limiting behavior: as the scene grows, the expansion frequency decreases significantly. This indicates that the pool rapidly accumulates the necessary semantic diversity and then stabilizes.
>
> We provide additional statistics below across multiple relatively large scenes (from Replica, TUM, ScanNet, and our in-the-wild data). The results show that although early frames trigger several expansions, the rate slows dramatically as more scene content is observed:
>
> | Scene | Total Frames Number | Expansion Frame IDs | Size Sequence | Final Size |
> |:---------:|:---:|:---:|:---:|:---:|
> | Replica Room 0 | 2000 | 16, 22, 118 | 25 $\rightarrow$ 50 $\rightarrow$ 100 $\rightarrow$ 200 | 200 |
> | TUM long office | 2585 | 2, 5, 180 | 25 $\rightarrow$ 50 $\rightarrow$ 100 $\rightarrow$ 200 | 200 |
> | ScanNet 0000 | 5578 | 119, 347, 645, 4292 | 25 $\rightarrow$ 50 $\rightarrow$ 100 $\rightarrow$ 200 $\rightarrow$ 400 | 400 |
> | In-the-wild data | 8501 | 168, 1041, 4569 | 25 $\rightarrow$ 50 $\rightarrow$ 100 $\rightarrow$ 200 | 200 |
>
> This behavior aligns with the design of the pool: once major semantic variations have been captured, incoming features tend to match existing slots rather than triggering new expansions.
>
> **Q5: Identification of new objects.**
>
> **A5:** Yes. An object is considered "appearing for the first time" when no valid correspondences can be established between its current semantic region and any previously observed keyframes. This procedure incurs negligible additional overhead, yet reliably detects first-appearance objects for the purpose of semantic stability guidance.
>
> **Q6: Discussion on system's performance in dynamic environments.**
>
> **A6:** Good comment! Highly dynamic environments are indeed challenging, and while our work focuses primarily on static scenes, the proposed semantic mechanisms provide promising foundations for handling dynamic ones.
>
> Current dynamic SLAM approaches [ref3–6] typically rely on 2D mask extraction to detect moving objects and then remove those regions during mapping. In contrast, our system introduces two properties that could naturally extend to dynamic settings:
>
> + **3D-level moving-object extraction:** Because each Gaussian is associated with a semantic feature from the pool, moving objects can be filtered directly in 3D by removing or isolating Gaussians assigned to dynamic semantic categories, rather than relying solely on 2D segmentation.
> + **Cross-frame semantic consistency for tracking dynamic objects:** Our consistency mechanism establishes correspondences across frames. This can help track and maintain coherent semantics for moving objects, aligning with recent trends in 3D dynamic-object reconstruction [ref6].
>
> While we have not explicitly targeted dynamic SLAM, these properties suggest that our approach could handle moderate dynamics and may serve as a strong basis for extending semantic-guided SLAM into dynamic environments. We consider this a promising direction for future work.
>
> [ref3] DG-SLAM: Robust Dynamic Gaussian Splatting SLAM with Hybrid Pose Optimization. NeurIPS 2024.
>
> [ref4] WildGS-SLAM: Monocular Gaussian Splatting SLAM in Dynamic Environments. CVPR 2025.
>
> [ref5] DyGS-SLAM: Real-Time Accurate Localization and Gaussian Reconstruction for Dynamic Scenes. ICCV 2025.
>
> [ref6] 4D Gaussian Splatting SLAM. ICCV 2025.

---

> > ### Comment · Reviewer_zwtS · 2025-11-28
> >
> > Thanks for authors' detailed response and the additional experiments regarding the baselines. Most of my concerns have been sufficiently addressed.

---

### Official Review · Reviewer_21bL · 2025-10-31

**Soundness:** 3
**Presentation:** 3
**Contribution:** 2
**Rating:** 4
**Confidence:** 4

**Summary:**

The paper introduces an open-set semantic 3D Gaussian Splatting (3DGS) framework designed to enhance scene reconstruction and semantic segmentation under open-set and real-time conditions. The proposed method integrates semantic understanding into the Gaussian Splatting pipeline by associating per-Gaussian semantic embeddings and enabling online label adaptation. The system claims to handle unknown categories and domain shifts, and to improve both reconstruction quality and scene-level segmentation.

Experiments are conducted mainly on the Replica dataset, showing improved reconstruction quality and semantic accuracy compared to several baseline methods. The results demonstrate promising performance in synthetic indoor environments, suggesting potential for robust open-world mapping.

**Strengths:**

1. Good presentation and writing quality.
The paper is clearly structured, logically consistent, and easy to follow. Figures are visually clean and the methodology is well explained.

2. Strong results on benchmark datasets.
The proposed approach achieves good reconstruction and segmentation results on Replica and other small-scale datasets, showing the method’s effectiveness in controlled settings.

3. Direct and effective idea.
The proposed framework is conceptually straightforward yet functional. It extends 3D Gaussian Splatting toward semantic understanding and contributes a practical perspective to open-set semantic reconstruction.

**Weaknesses:**

1. Overreliance on synthetic data.
Using Replica as the main evaluation dataset is limiting. Since Replica is synthetic and lacks real-world sensor noise, it is less meaningful for evaluating SLAM or localization robustness. The paper would be stronger with experiments on real-world semantic datasets such as ScanNet++, SemanticKITTI.

2. Missing comparisons with recent baselines.
Several recent 3DGS-based SLAM and semantic mapping methods (e.g., MonoGS, S3PO-GS, SEGS-SLAM) are not included in comparisons. Without these baselines, it is difficult to gauge the real advancement of the proposed approach relative to the current state of the art.

3. Limited improvement from semantics to pose estimation.
While the paper introduces semantic components, the pose estimation accuracy remains very close to previous methods, suggesting that semantics may not significantly contribute to the geometric optimization process.

**Questions:**

The questions are the same as weakness. This paper presents a solid and well-written approach that meaningfully extends semantic understanding into 3D Gaussian Splatting. However, the experimental evaluation is too narrow and overly dependent on synthetic data, which weakens claims of real-world robustness and open-set generalization. Including stronger baselines and more diverse datasets would substantially improve the paper’s credibility.
Typos:
1. In Figure 2, the label “Ground Truth RGB Frame” appears misplaced — please verify and correct this annotation.

---

> ### Author Response · Authors · 2025-11-22
> **Point-to-Point Response to Reviewer 21bL**
>
> We thank reviewer 21bL for the valuable time and constructive feedback. We provide point-to-point response below.
>
> **Q1: "Using Replica as the main evaluation dataset is limiting."**
>
> **A1:** Sorry for the misunderstanding. We would like to clarify that our evaluation protocol strictly follows the prevailing practice in the semantic SLAM literature, especially SGS-SLAM^ [58].
>
> Across existing semantic SLAM works (*e.g.*, SNI-SLAM [57], SGS-SLAM [58], GS$^3$LAM [61]), evaluations fall into two categories: tracking and appearance mapping (ATE RMSE, PSNR, SSIM, LPIPS), typically evaluated on Replica, ScanNet, and TUM; and semantic classification (mIoU), for which Replica is the only used benchmark. The datasets mentioned (*i.e.*, ScanNet++ and SemanticKITTI) are valuable but are not part of the standard evaluation protocol in semantic SLAM.
>
> For completeness, our tracking and appearance mapping results on ScanNet and TUM are already provided in Tables 15 and 16. To further address your concern, we conduct a closed-set semantic classification experiment on ScanNet. Following GS$^3$LAM [61], we use scene0059_00, which is also designated as an example in their official GitHub repository.
>
> | Methods | mIoU (%) |
> |:---:|:---:|
> | GS$^3$LAM | 18.62 |
> | **Ours** | **53.27** |
>
> Furthermore, as shown in Table 3, we provide 2D segmentation and 3D localization results on our in-the-wild data, achieving more than 7.2% mIoU improvement.
>
> We agree that incorporating datasets such as ScanNet++ and SemanticKITTI would further highlight scalability and generalization, and we consider these directions valuable for future work.
>
> ^ SGS-SLAM noted that they "only show four scenes because previous NeRF-based semantic models only reported results on these scenes."
>
> **Q2: Regarding "missing comparisons with recent baselines".**
>
> **A2:** Thanks for the suggestion. Our focus is not on proposing a new SLAM system, but rather on introducing a representation that enhances the semantic reasoning capabilities of any GS-based SLAM pipeline. Our approach is fundamentally complementary: rather than replacing existing SLAM frameworks, it augments them. Therefore, comparing our method as if it were itself a full SLAM system can lead to an unfair characterization of "missing baselines."
>
> However, we agree that it is valuable to demonstrate how our semantic representation performs when plugged into recent SOTA GS-based SLAM frameworks. We conduct comparisons on Replica against several recent baselines, including MonoGS [21], LoopSplat [19], S3PO-GS [ref1], and SEGS-SLAM [ref2].
>
> | Methods | ATE RMSE $\downarrow$ | PSNR $\uparrow$ | SSIM $\uparrow$ | LPIPS $\downarrow$ |
> |:----:|:---:|:---:|:---:|:---:|
> | MonoGS [21] [CVPR24] | 0.79 | 37.50 | 0.960 | 0.070 |
> | **MonoGS + Ours** | **0.55** | **38.39** | **0.976** | **0.049** |
> | LoopSplat^ [19] [3DV25] | 0.26 | 36.63 | 0.985 | 0.112 |
> | **LoopSplat + Ours** | **0.23** | **38.05** | **0.989** | **0.051** |
> | S3PO-GS^^ [ref1] [ICCV25] | 2.26 | 32.71 | 0.943 | 0.138 |
> | **S3PO-GS + Ours** | **0.97** | **35.23** | **0.961** | **0.070** |
> | SEGS-SLAM [ref2] [ICCV25] | 0.43 | 39.42 | 0.975 | 0.021 |
> | **SEGS-SLAM + Ours** | **0.38** | **39.88** | **0.983** | **0.020** |
>
> ^ We already included LoopSplat (3DV 2025) in our main text; it represents a strong recent baseline.
>
> ^^ S3PO-GS is designed for open-world scenarios; the Replica results shown here come from our reproduction.
>
> [ref1] Outdoor Monocular SLAM with Global Scale-Consistent 3D Gaussian Pointmaps. ICCV 2025.
>
> [ref2] SEGS-SLAM: Structure-enhanced 3D Gaussian Splatting SLAM with Appearance Embedding. ICCV 2025.
>
> **Q3: Discussion on the improvement to pose estimation.**
>
> **A3:** Please bear with our clarification below. The camera pose estimation alone does not fully capture the effectiveness of geometric optimization. Our semantic representation is designed to improve the underlying geometric reconstruction and optimization, which is not always directly reflected by pose metrics such as ATE RMSE.
>
> **First, semantics substantially improve geometric optimization.** As shown in Table 1c, incorporating our semantic representation leads to a **43.1%–52.8%** reduction in Depth L1, indicating a major improvement in geometric fidelity compared to baselines. Fig. 4 visually illustrates that our method corrects inaccurate or unstable geometric optimization.
>
> **Second, pose estimation is not our primary focus, yet still improves meaningfully.** While pose tracking is not the central contribution of our work, our method nonetheless yields **11.5%–19.4%** improvement in ATE RMSE, which is a substantial gain given that (1) modern GS-based SLAM systems already have strong tracking performance (*e.g.*, S3PO-GS [ref1] and SEGS-SLAM [ref2] both gain no improvement against previous works) and (2) pose estimation is an indirect beneficiary of our semantic-guided geometric optimization.
>
> **Q4: Typo.**
>
> **A4:** Thank you for your careful review. We have corrected it.

---

> > ### Comment · Reviewer_21bL · 2025-11-26
> >
> > Thank you for the author's response.
> >
> > Regarding A1, "Strictly following the datasets evaluated by previous SLAM systems" is not a reasonable justification. The reasons are as follows:
> >
> > 1. NeRF-based SLAM methods (IMAP, NICE-SLAM) were indeed initially evaluated on these datasets, and Semantic NeRF/3DGS SLAM followed suit at the beginning. However, as the field has evolved, both NeRF-based and 3DGS-based SLAM methods have gradually expanded to newer datasets or added more scenes to existing ones, such as ScanNet++, Waymo, etc. ***This is essentially because the original datasets have become too simple, and the metrics are already sufficiently high such that further improvements are not meaningful***. For example, the ATE RMSE improvements of 0.xx cm on the synthetic Replica dataset and PSNR improvements on top of an already high baseline of 36+ in your paper do not convincingly demonstrate the effectiveness of your method.
> >
> > 2. Your paper claims advantages in open-set scenarios and everyday device convenience. Under this premise, using only Replica is indeed insufficient to support your claims.
> >
> > 3. ***Even setting these concerns aside, the results on Replica show no significant advantage.*** I understand your paper presents a plugin rather than a full system, showing some improvement over non-semantic baselines on Replica, TUM, and ScanNet. ***However, why not report evaluation results from semantic SLAM methods in the paper, such as Open-GS SLAM's tracking and PSNR on Replica (0.16, 39.49)? Plugin-based methods inherently introduce additional modules, so improvements over the original baseline are expected.*** However, to truly assess whether this plugin is worthwhile, comparison with similar methods would be more fair.
> >
> > Regarding A2, I appreciate the author's hard work and have no further question.
> >
> > Regarding A3, I disagree with the statement "pose estimation is not our primary focus, yet still improves meaningfully" for the same reasons outlined in my response to A1. Moreover, if pose estimation is not your primary focus, then claiming this as a SLAM in title and paper is a little inappropriate.
> >
> > ***I primarily have concerns regarding the experimental section, as I worry it may not adequately support the paper's claims, all of which have been outlined above.*** Additionally, the statement "Splat-SLAM [50] is not yet accepted to any conference or journal" (in your response to reviewer r2wC) is inaccurate: https://openaccess.thecvf.com/content/CVPR2025W/VOCVALC/html/Sandstrom_Splat-SLAM_Globally_Optimized_RGB-only_SLAM_with_3D_Gaussians_CVPRW_2025_paper.html
> >
> > 1. Sucar, E., Liu, S., Ortiz, J., & Davison, A. J. (2021). imap: Implicit mapping and positioning in real-time. In Proceedings of the IEEE/CVF international conference on computer vision (pp. 6229-6238).
> > 2. Zhu, Z., Peng, S., Larsson, V., Xu, W., Bao, H., Cui, Z., ... & Pollefeys, M. (2022). Nice-slam: Neural implicit scalable encoding for slam. In Proceedings of the IEEE/CVF conference on computer vision and pattern recognition (pp. 12786-12796).
> > 3. Yang, D., Gao, Y., Wang, X., Yue, Y., Yang, Y., & Fu, M. (2025). OpenGS-SLAM: Open-Set Dense Semantic SLAM with 3D Gaussian Splatting for Object-Level Scene Understanding. arXiv preprint arXiv:2503.01646.

---

> > > ### Author Response · Authors · 2025-11-30
> > > **Thanks for your response and further discussion (1/2)**
> > >
> > > Thank you so much for your response.
> > >
> > > **Q1: "'Strictly following the datasets evaluated by previous SLAM systems' is not a reasonable justification."**
> > >
> > > **A1:** Thank you for your feedback. We understand your concern regarding the need for more experimental results on real-world data. We address your concerns below.
> > >
> > > > **Q1.1: "This is essentially because the original datasets have become too simple, and the metrics are already sufficiently high such that further improvements are not meaningful."**
> > >
> > > **A1.1:** Sorry for the confusion. As mentioned, we have already included results on real-world datasets such as ScanNet and TUM in Tables 17 and 18. We would like to clarify that, to the best of our knowledge, ScanNet++ is NOT yet widely adopted across many research works (*e.g.*, MonoGS [44], Photo-SLAM [47], GS-ICP SLAM [52], SEGS-SLAM [56]), while Waymo has NOT been used. However, to better address your concerns, we provide additional results on ScanNet++.
> > >
> > > Novel View:
> > > | Methods | PSNR $\uparrow$ | SSIM $\uparrow$ | LPIPS $\downarrow$ |
> > > | :---: | :---: | :---: | :---: |
> > > | SplaTAM | 24.41 | 0.880 | 0.240 |
> > > | **SplaTAM + Ours** | **25.78** | **0.931** | **0.162** |
> > >
> > > Training View:
> > > | Methods | PSNR $\uparrow$ | SSIM $\uparrow$ | LPIPS $\downarrow$ |
> > > | :---: | :---: | :---: | :---: |
> > > | SplaTAM | 27.98 | 0.940 | 0.120 |
> > > | **SplaTAM + Ours** | **29.91** | **0.968** | **0.080** |
> > >
> > > We further provide experimental results on three in-the-wild scenes, which are visually presented in Figure 8:
> > >
> > > Scene 1：
> > > | Methods | PSNR $\uparrow$ | SSIM $\uparrow$ | LPIPS $\downarrow$ |
> > > | :---: | :---: | :---: | :---: |
> > > | SplaTAM | 22.75 | 0.864 | 0.231 |
> > > | **SplaTAM + Ours** | **25.51** | **0.915** | **0.164** |
> > >
> > > Scene 2：
> > > | Methods | PSNR $\uparrow$ | SSIM $\uparrow$ | LPIPS $\downarrow$ |
> > > | :---: | :---: | :---: | :---: |
> > > | SplaTAM | 21.30 | 0.812 | 0.295 |
> > > | **SplaTAM + Ours** | **24.18** | **0.888** | **0.226** |
> > >
> > > Scene 3：
> > > | Methods | PSNR $\uparrow$ | SSIM $\uparrow$ | LPIPS $\downarrow$ |
> > > | :---: | :---: | :---: | :---: |
> > > | SplaTAM | 17.94 | 0.786 | 0.366 |
> > > | **SplaTAM + Ours** | **20.84** | **0.848** | **0.296** |
> > >
> > > > **Q1.2: Our advantages in open-set scenarios and everyday device convenience.**
> > >
> > > **A1.2:** As shown in Figures 1, 8, and 14, we present visual results on real-world data that illustrate the performance of our method in open-set scenarios. We believe these visualizations, together with the quantitative results provided in **A1.1** and Tables 17 and 18, adequately demonstrate the robustness of our method across a variety of real-world scenarios.
> > >
> > > > **Q1.3: The results on Replica show no significant advantage.**
> > >
> > > **A1.3:** We have conducted comparative experiments with GS$^3$LAM [63], which is based on SplaTAM, and *our approach shows greater performance improvements*, validating its effectiveness. Regarding OpenGS-SLAM, we have not included it in our comparison due to the lack of a publicly available implementation, which prevents us from ensuring a fair and reproducible evaluation. We will clarify this point in the final version of our paper.

---

> > > > ### Author Response · Authors · 2025-11-30
> > > > **Thanks for your response and further discussion (2/2)**
> > > >
> > > > **Q2: More discussion on the improvement to pose estimation.**
> > > >
> > > >
> > > > > **Q2.1: "I disagree with the statement ... for the same reasons outlined in my response to A1."**
> > > >
> > > > **A2.1:** We assert that *tracking performance is not a weakness of our approach, and we have made meaningful contributions to the semantic SLAM field*. **First**, on Replica, as previously discussed, the performance improvement of **11.5%–19.4%** is statistically significant compared to other state-of-the-art methods. For instance, two recent works accepted by ICCV 2025, S3PO-GS [55] and SEGS-SLAM [56], do not show improvements in tracking accuracy (as demonstrated by the table in our first response). Additionally, within the semantic SLAM domain, our approach delivers a more substantial performance gain than other methods, such as SGS-SLAM [60] (which shows a 9.25% improvement) and GS$^3$LAM [63] (which actually shows a -2.78% decrease). **Second**, on ScanNet, our method demonstrates greater robustness, achieving a more significant improvement over the SplaTAM-based GS$^3$LAM (10.27 vs. 11.96). **Third**, on TUM, while other semantic SLAM approaches, such as SGS-SLAM and GS$^3$LAM, fail to run, our method consistently improves baseline performance under the same conditions.
> > > >
> > > > > **Q2.2: "If pose estimation is not your primary focus, then claiming this as a SLAM in title and paper is a little inappropriate."**
> > > >
> > > > **A2.2:** We respectfully disagree. **First**, we remain strongly focused on tracking performance and have introduced an additional semantic loss to enhance the stability of camera pose estimation. Experimental results confirm that this improvement has led to better tracking accuracy (Table 1b). **Second**, our framework is built on complete SLAM systems, integrating both tracking and mapping components. Removing the tracking module would make systems inoperable, which justifies our classification of the method as a full SLAM approach. **Third**, it is important to emphasize that our method is specifically designed to address core challenges in SLAM, such as handling uncertain scene scales and maintaining semantic consistency throughout the process. **Lastly**, we note that many well-established SLAM systems (*e.g.*, iMAP [36], NICE-SLAM [37], SplaTAM [20], Photo-SLAM [47], SEGS-SLAM [56]), including semantic SLAM methods (*e.g.*, SNI-SLAM [59], SGS-SLAM [60], GS$^3$LAM [63], OpenGS-SLAM [65]), do not prioritize tracking as their primary focus, yet they are still recognized as mature, complete, and influential works in the field.
> > > >
> > > > **Q3: Statement on Splat-SLAM.**
> > > >
> > > > **A3:** Sorry for the confusion. We believe that excluding _workshop_ papers from our comparisons should not be viewed as a limitation. However, to further enhance the robustness of our work, we will consider incorporating such comparisons in future versions.

---

### Author Response · Authors · 2025-11-30
**Rebuttal Summary**

We would like to express our sincere gratitude to the AC for their tremendous efforts in managing the unexpected events. We also deeply appreciate the constructive comments from all the reviewers, which have significantly improved the quality of our paper. We believe that our responses have adequately addressed the reviewers' concerns, and we summarize the key points below:

1. **Extra comparisons with more baselines.** To better address reviewer **21bL** and **r2wC**'s concerns, we extend experiments on more recent baselines (*i.e.*, S3PO-GS, SEGS-SLAM). Results demonstrate a robust performance improvement (*e.g.*, over **+11.6%** in ATE RMSE and over **+0.46** in PSNR), strongly indicating the effectiveness of our method across diverse baselines.
2. **Extra evaluations on real-world data.** While our paper already includes results on real-world data (Tables 17, 18 & Figs. 1, 8, 14), we expand our experiments to address reviewer **21bL**'s concerns as follows:
+ We perform closed-set evaluations on ScanNet, comparing against GS$^3$LAM (**+34.65%** in mIoU).
+ We include experiments on ScanNet++ (over **+1.37** in PSNR on novel views) and our in-the-wild data (over **+2.90** in PSNR).
3. **Further clarifications.** Regarding reviewers' questions, we express the following clarifications:
+ **Improvements in camera pose estimations.** Our method achieves substantial improvements in camera pose estimation compared to semantic SLAM baselines (*e.g.*, **+11.5%** *vs.* GS$^3$LAM's -2.78%). Additionally, our method demonstrates robustness on real-world data, where other semantic SLAM frameworks fail to run.
+ **Usage of fundamental models.** Our method is not dependent on any specific semantic extraction model. In our experiments, we use SAM+CLIP to ensure a fair and direct comparison with prior works such as LERF and GOI. This flexible design makes our approach future-proof: as stronger or less biased foundation models become available, they can be easily integrated into our pipeline, further enhancing semantic representation quality without altering the core method.

We appreciate the reviewers' thoughtful feedback, and we have carefully addressed their concerns. Reviewer **21bL** has no further question for extra comparisons with more baselines and raises some points to discuss, and we have provided responses to them. Reviewer **zwtS**'s concerns have been addressed. Reviewer **r2wC** has no further questions and agrees to raise points. Reviewer  **fyiJ** has no question.

We again thank the reviewers for their valuable feedback and the AC for their hard work in managing this process.

Sincerely yours,

Authors.

---

### Meta-Review · Area_Chair_uFtG · 2026-01-09

**Summary:**

Before the rebuttal and discussion phase, the initial 4 reviews are mixed. All reviewers are satisfied with the quality of this paper and most of them admire the novelty and idea.

The most important concern, raised by multiple reviewers, is the missing SOTA competitors. During the discussion phase, the authors have added the compared experiments and one of the reviewers is satisfied with the results.

Although there are some other concerns such as over-reliance on synthetic data and limited novelty of using CLIP and DINOv2, the performance gain seems promising at least.

Overall, the strengths outweigh the weaknesses. I recommend it for acceptance.

**Reviewer Concerns:**

The comparisons with the recent SOTA models.

**Reviewer Scores:**

Most reviewers may raise their rating to positive scores due to the additional efforts made by the authors during the rebuttal phase. In particular, one reviewer claimed that he/she were willing to raise the rating from a borderline reject.

---

### Decision · Program_Chairs · 2026-01-26

Accept (Poster)